# Wasserstein Proximal Policy Gradient: Implicit Policies, Entropy Regularization and Linear Convergence

## Abstract

We revisit Wasserstein Proximal Policy Gradient (WPPG) for continuous control in infinite-horizon discounted reinforcement learning. By projecting the iterate of Wasserstein proximal gradient onto a parametric policy family with respect to the Wasserstein distance, we derive a new WPPG update that eliminates the need for policy densities or score functions. This makes our method directly applicable to implicit stochastic policies. We prove a linear convergence rate for the WPPG iterate under entropy regularization and Talagrand's transport-entropy inequality on the policy class, for both exact and approximate value function estimates. Empirically, our algorithm is simple to implement and achieves competitive performance on standard benchmarks.

## 1 Introduction

Reinforcement learning (RL) has become a powerful paradigm for solving complex sequential decision-making problems, powering landmark achievements from superhuman performance in strategic games (Silver et al., 2016; 2017) and advanced robotic control (Levine et al., 2016) to the training of Large Language Models (Guo et al., 2025). At the heart of many of these successes are policy gradient (PG) methods (Williams, 1992; Sutton et al., 1999), which iteratively update a parameterized policy to maximize expected rewards.

The geometry underlying policy updates plays an important role in the learning process. Standard policy gradient methods use the Euclidean geometry of parameter space, while the natural policy gradient (Kakade, 2001) and trust-region methods such as TRPO (Schulman et al., 2017a) and PPO (Schulman et al., 2017b) instead exploit the information geometry of policies via Kullback–Leibler (KL) divergence. These methods are supported by a growing body of analysis, with recent results establishing fast global convergence rates in finite action spaces (Agarwal et al., 2020; Lan, 2021; Xiao, 2022; Cen et al., 2022; Bhandari & Russo, 2024).

Recent work explores an alternative paradigm that formulates policy optimization in distribution space under the Wasserstein metric. This perspective builds on the theory of gradient flows in probability spaces (Zhang et al., 2018; Moskovitz et al., 2021; Ziesche & Rozo, 2023; Pfau et al., 2025). In contrast to KL-based methods, which treat actions as independent categories, Wasserstein-based approaches inherently respect the geometry of the action space, capturing meaningful notions of proximity between actions (Pacchiano et al., 2020; Moskovitz et al., 2021; Song et al., 2023). The resulting stochastic policy updates rely on the gradient of the action-value function with respect to the action, drawing a close connection to deterministic policy gradients (Pfau et al., 2025).

While Wasserstein policy optimization offers a compelling alternative to KL-based methods, its theoretical foundations are far less developed. Using the JKO framework (Jordan et al., 1998) for Wasserstein gradient flows, Zhang et al. (2018) established asymptotic convergence of the entropy-regularized problem when the policy distribution is approximated by particles. For finite action spaces, Song et al. (2023) proved global convergence as the Wasserstein penalty coefficient vanishes. Beyond these results, however, convergence guarantees in the more general setting of continuous action spaces—particularly for parametric policies beyond particle approximations (e.g., mixtures of Gaussians)—remain, to the best of our knowledge, an open question.

In this paper, we introduce a version of the Wasserstein policy optimization, which we term Wasserstein Proximal Policy Gradient (WPPG). Our approach projects the theoretical Wasserstein gradient flow onto the parametric policy family, using the Wasserstein metric as the projection criterion. This

contrasts with Pfau et al. (2025), which effectively relies on the KL divergence to project the flow onto the parametric policy manifold. Our main findings are as follows.

- Our WPPG update introduces a new scheme for optimizing stochastic policies. Unlike most existing approaches, it relies solely on the gradient of the action–value function with respect to the action, without requiring access to the policy distribution's (log-)density or score function. This enables a novel approach to policy optimization with *implicit policies* (Tang & Agrawal, 2018). Empirically, the resulting algorithm is simple to implement and demonstrates competitive performance on standard continuous-control benchmarks.

- We establish a linear convergence rate of WPPG for the entropy-regularized problem, under some regularity and approximate realizability assumptions that can be satisfied by certain implicit policy classes. Our analysis applies to both the exact and approximate value function estimation.

## 1.1 RELATED WORKS

**On Wasserstein policy update**  Zhang et al. (2018) formulate continuous-time entropy-regularized policy optimization as a gradient flow of an energy functional, and derive its discrete-time counterpart via the JKO scheme (Jordan et al., 1998), parameterizing policies with particles or energy-based models. Related gradient-flow perspectives also appear in Richemond & Maginnis (2018), and in Ziesche & Rozo (2023) for mixtures of Gaussian policies. Moskovitz et al. (2021) introduce a Wasserstein trust-region policy update and develop efficient kernel-based estimators. Song et al. (2023) study Wasserstein and Sinkhorn trust-region updates in finite action spaces, with an extension to one-dimensional continuous control, and derive closed-form policy updates via duality. More recently, Pfau et al. (2025) propose a Wasserstein gradient flow–inspired update by projecting the Wasserstein gradient flow on parametric manifold using KL divergence.

Most of the above-mentioned Wasserstein policy updates rely on the (log-)density of the policy distribution (or probability mass function for finite action spaces) and/or its score functions, with the exception of Moskovitz et al. (2021), whose kernel-based method instead requires gradients of the kernel for implicit models. In particular, our update, like those of Zhang et al. (2018); Pfau et al. (2025), depends on the gradient of the action-value function with respect to the action. However, unlike both of these two approaches, our method does not rely on the (log-)density of the policy distribution. This is achieved by handling the entropy term through Gaussian noise injection, rather than working directly with the density as in Zhang et al. (2018), and by projecting under the Wasserstein metric instead of the KL divergence, as in Pfau et al. (2025).

We remark that other related approaches leverage Wasserstein geometry in different ways. For instance, Pacchiano et al. (2020) compare policies in latent behavior spaces but do not focus on explicit policy updates, while Abdullah et al. (2019) use Wasserstein distances for robust uncertainty modeling, treating Wasserstein as a constraint on the transition dynamics of the environment rather than as a tool for defining learning dynamics.

**On convergence analysis**  Zhang et al. (2018) establish asymptotic convergence of Wasserstein policy optimization based on the JKO scheme when policies are approximated by particles. Song et al. (2023) proves linear convergence analysis for Wasserstein trust-region optimization on finite action spaces, but requires vanishing Wasserstein penalty coefficients. Our convergence proof strategy parallels KL-based analyses such as Lan (2021) for mirror policy gradient in finite action spaces. However, instead of relying on KL-specific tools (e.g., the three-point identity), we develop new analyses tailored to Wasserstein geometry and adopt different assumptions on the problem.

## 2 PRELIMINARIES

### 2.1 REINFORCEMENT LEARNING

**Markov decision processes.**  We consider an infinite-horizon discounted MDP $\mathcal{M} = (\mathcal{S}, \mathcal{A}, P, r, \rho, \gamma)$, where $\mathcal{S}$ is the state space, $\mathcal{A}$ is the action space, $\gamma \in (0, 1)$ the discount factor, $\mathbb{P}(\cdot \mid s, a)$ the transition kernel, $r : \mathcal{S} \times \mathcal{A} \to \mathbb{R}$ the reward function, and $\rho$ the initial-state distribution. A policy is a Markov kernel $\pi : \mathcal{S} \to \mathcal{P}(\mathcal{A})$ so that $\pi(\cdot \mid s)$ is a probability distribution on $\mathcal{A}$ for each $s \in \mathcal{S}$. Our results apply to the case where $\mathcal{A}$ is a general metric space.

For a policy $\pi$, the value and action-value (Q) functions are

$$V^\pi(s) := \mathop{\mathbb{E}}_{\substack{a_t \sim \pi(\cdot|s_t) \\ s_{t+1} \sim P(\cdot|s_t, a_t)}} \left[ \sum_{t=0}^\infty \gamma^t r(s_t, a_t) \Big| s_0 = s \right],$$

$$Q^\pi(s,a) := \mathop{\mathbb{E}}_{\substack{a_t \sim \pi(\cdot|s_t) \\ s_{t+1} \sim P(\cdot|s_t, a_t)}} \left[ \sum_{t=0}^\infty \gamma^t r(s_t, a_t) \Big| s_0 = s, \ a_0 = a \right].$$

We have the relationship that $V^\pi(s) = \mathbb{E}_{a \sim \pi(\cdot|s)}[Q^\pi(a,s)]$. The advantage function is $A^\pi(s,a) := Q^\pi(s,a) - V^\pi(s)$. Given an initial distribution $\rho$, the performance (expected return) is $J_\rho(\pi) := \mathbb{E}_{s_0 \sim \rho}[V^\pi(s_0)]$. The discounted state visitation probability is defined as $d_\rho^\pi(s) := (1-\gamma) \sum_{t=0}^\infty \gamma^t \mathbb{P}_\pi(s_t = s \mid s_0 \sim \rho)$, with $\sum_s d_\rho^\pi(s) = 1$.

**Entropy regularization.** Entropy regularization is widely used in reinforcement learning to prevent premature policy collapse and encourage exploration. It smooths the optimization landscape, reduces gradient variance, and promotes robust, generalizable strategies. These benefits make it a standard component in modern algorithms like TRPO, PPO, and SAC. Define the negative entropy of a policy $\pi$ at state $s$ as $\mathsf{H}^\pi(s) := \int_{a \in \mathcal{A}} \pi(a|s) \log \pi(a|s) \mathrm{d}a$. Define the modified reward $r_\tau(s,a) := r(s,a) - \tau \log \pi(a \mid s)$. Then $V_\tau^\pi$ is the value of $\pi$ under $r_\tau$. The corresponding *soft-V and soft-Q* functions satisfy the soft Bellman recursion

$$V_\tau^\pi(s) := \mathbb{E}^\pi \left[ \sum_{t=0}^\infty \gamma^t \big( r(s_t, a_t) - \tau \mathsf{H}^\pi(s_t) \big) \Big| s_0 = s \right], \tag{1}$$

$$Q_\tau^\pi(s,a) := r(s,a) - \tau \mathsf{H}^\pi(s) + \gamma \mathbb{E}\left[V_\tau^\pi(s') \mid s,a\right], \qquad V_\tau^\pi(s) = \mathbb{E}_{a \sim \pi(\cdot|s)}\left[Q_\tau^\pi(s,a)\right].$$

where the temperature $\tau \geq 0$ trades off exploitation and exploration.

**Policy Gradient** Policy gradient methods form a core class of reinforcement learning algorithms, particularly effective in high-dimensional or continuous settings. They optimize parameterized policies $\pi = \pi_\theta$ directly through gradient ascent on the parameter $\theta$. The geometry underlying these updates significantly influences their behavior. Vanilla policy gradient uses Euclidean steps in parameter space, updating parameters via simple gradient ascent; natural policy gradient incorporates information geometry by equipping the parameter space with the Fisher information metric, yielding updates that are invariant to smooth reparameterizations and closely related to trust-region methods; Mirror or proximal policy gradient methods operate directly in the space of probability distributions, updating policies through mirror ascent with respect to a Bregman divergence such as KL.

## 2.2 Wasserstein Proximal Gradient

We next introduce a proximal gradient scheme formulated in the 2-Wasserstein geometry. To stay consistent with our reinforcement learning framework, we present it as a maximization problem and emphasize the underlying intuition; for a rigorous mathematical treatment, we refer the reader to Ambrosio et al. (2008); Santambrogio (2015).

Consider the problem of maximizing an entropy-regularized functional

$$\max_q F_0(q) - \tau \mathsf{H}(q), \tag{2}$$

where $\mathsf{H}(q) = \mathbb{E}_q[\log q]$. At iteration $k$, consider the Wasserstein proximal gradient update defined as

$$q_{k+1} = \operatorname{argmax}_q \left\langle \tfrac{\delta F_0}{\delta q}[q_k], \ q \right\rangle - \tau \mathsf{H}(q) - \frac{1}{2\eta} \mathsf{W}_2^2(q, q_k), \tag{3}$$

where $\frac{\delta F}{\delta q}[q_k]$ denotes the first variation of $F$ at $q_k$, and $\mathsf{W}_2$ denotes the 2-Wasserstein metric, and we write $\langle f, \mu \rangle := \int f \, \mathrm{d}\mu$ for the function–measure pairing. For simplicity, we take the transport cost on the action space $\mathcal{A}$ to be the norm $\|\cdot - \cdot\|$, although our results extend to more general cost functions. Analogous to the Euclidean setting, the Wasserstein proximal gradient update can be interpreted as selecting, among all candidates close to the current iterate, the one that most improves the local linearization of the objective.

The continuous-time limit of the Wasserstein proximal gradient scheme above is given by the Fokker-Planck equation

$$\partial_t q_t(a) = -\operatorname{div} \cdot \left( q_t(a) \nabla \tfrac{\delta F_0}{\delta q}[q_t](a) \right) + \tau \Delta q_t(a),$$

where the diffusion (Laplacian) term arises from the entropy regularization and the fact that

$$-\text{div} \cdot \left( q(a) \nabla \tfrac{\delta \mathsf{H}}{\delta q}(a) \right) = \text{div} \cdot \left( q(a) \nabla \log q(a) \right) = \Delta q(a).$$

Using Euler-Maruyama discretization, the entropy-induced diffusion can be implemented by injecting Gaussian noise at each iteration. Specifically, a sample $a \sim q_k$ is updated as

$$a \mapsto a + \eta \nabla \tfrac{\delta F_0}{\delta q}[q_k](a) + \sqrt{2\tau\eta}\,\xi, \quad \xi \sim \mathcal{N}(0, I). \tag{4}$$

This update corresponds to one step of noisy Wasserstein gradient ascent, where the drift term pushes samples toward regions of higher value of $\tfrac{\delta F_0}{\delta q}[q_k](a)$ and the diffusion term encourages exploration with strength controlled by $\tau$.

## 3 WASSERSTEIN PROXIMAL POLICY GRADIENT

We now apply the Wasserstein proximal gradient scheme described in Section 2.2 to the reinforcement learning setting. Define $\tilde{V}_\tau^\pi(s) = V_\tau^\pi(s) + \tau\mathsf{H}^\pi(s)$. We rewrite the soft value function (1) as

$$\tilde{V}_\tau^\pi(s) - \tau\mathsf{H}^\pi(s), \tag{5}$$

which mirrors the structure of (2). For each state $s$, the first variation of $\tilde{V}_\tau^\pi(s)$ with respect to the policy $\pi$ is given by (see Lemma 3 in Appendix):

$$\frac{\delta \tilde{V}_\tau^\pi}{\delta \pi}(a|s) = \frac{1}{1-\gamma} d_{s_0}^\pi(s) Q_\tau^\pi(s, a).$$

Apply the Wasserstein proximal gradient update (3) to problem (5), we obtain the following update rule for the policy:

$$\pi_{k+1}(\cdot|s) = \text{argmax}_{q(\cdot|s) \in \Pi(s)} \langle Q_\tau^{\pi_k}(s, \cdot), q(\cdot|s) \rangle - \tau\mathsf{H}^q(s) - \frac{1}{2\eta} \mathsf{W}_2^2\big(q(\cdot|s), \pi_k(\cdot|s)\big), \quad \text{(WPPG)}$$

where $\Pi(s)$ denotes the policy class over which we optimize. Equivalently, this can be viewed as the solution to the following Wasserstein policy proximal gradient problem with Wasserstein metric weighted by the state visitation distribution:

$$\max_{\pi \in \Pi} \mathbb{E}_{s \sim d_{s_0}^{\pi_k}, a \sim \pi(\cdot|s)} \big[A_\tau^{\pi_k}(s, a)\big] - \frac{1}{2\eta} \mathbb{E}_{s \sim d_{s_0}^{\pi_k}} \big[\mathsf{W}_2^2\big(\pi(\cdot|s), \pi_k(\cdot|s)\big)\big],$$

where the advantage function $A_\tau^{\pi_k}(s, a) = Q_\tau^{\pi_k}(s, a) - V_\tau^{\pi_k}(s)$ replaces the action-value function $Q_\tau^{\pi_k}(s, a)$ since $V_\tau^{\pi_k}(s)$ does not depend on the optimization variable $\pi(\cdot|s)$. When $\tau = 0$, this reduces to the formulation considered in Song et al. (2023); Pfau et al. (2025). When the Wasserstein distance is replaced with Bregman's distance, (WPPG) becomes the mirror descent policy optimization in Lan (2021).

### 3.1 PRACTICAL ALGORITHM WITH IMPLICIT POLICIES

Suppose now that the (stochastic) policy class is parameterized as an implicit generative model

$$a = g_\theta(s, Z), \quad \theta \in \Theta,$$

where the latent variable $Z \sim \nu$ is drawn from an easy-to-sample distribution. In this parameterization, using the drift-diffusion update (4), WPPG at state $s$ corresponds to transporting each action sample $a = g_{\theta_k}(s, Z)$ to

$$\tilde{a}_s(Z, \xi) := g_{\theta_k}(s, Z) + \eta \nabla_a Q_\tau^{\pi_k}(s, g_{\theta_k}(s, Z)) + \sqrt{2\tau\eta}\,\xi, \quad \xi \sim \mathcal{N}(0, I).$$

Let $\tilde{\mu}_s$ be the distribution of $\tilde{a}_s$. Since the exact transported distribution generally does not lie in the parametric family $\{g_\theta(s, \cdot)_{\#}\nu : \theta \in \Theta\}$, we update the policy parameters by projection under the Wasserstein metric—in contrast to Pfau et al. (2025) in which information geometry is involved in the projection. Specifically, we first project the drift component onto the parametric family

$$\min_\theta \mathsf{W}_2^2\big(g_\theta(s, \cdot)_{\#}\nu, \tilde{\mu}_s\big). \tag{6}$$

When the step size $\eta$ is small, the shared-latent coupling

$$\big(g_\theta(s, Z), \tilde{a}_s(Z, \xi)\big), \qquad Z \sim \nu$$

induced by sharing the same latent variable $Z \sim \nu$, is first-order optimal for computing the Wasserstein distance (6). This follows from the geometry of optimal transport: the 2-Wasserstein distance between $q$ and its displacement $(\text{id} + v)_{\#}q$ along a vector field $\eta v$ is given by $\mathsf{W}_2^2(q, (\text{id} + \eta v)_{\#}q) = \eta^2 \mathbb{E}_{a \sim q}[\|v(a)\|^2] + o(\eta^2)$ (Ambrosio et al., 2008, Theorem 8.4.6); and the coupling $(a, a + \eta v(a))$ with $a \sim q$ achieves this value up to $o(\eta^2)$. Therefore, to update $\theta$, we can minimize the expected squared distance under the shared-latent coupling

$$\theta_{k+1} = \text{argmin}_\theta \, \mathbb{E}_{s \sim d^{\pi_k}, Z \sim \nu} \Big[\|g_\theta(s, Z) - g_{\theta_k}(s, Z) - \eta \nabla_a Q_\tau^{\pi_k}(s, g_{\theta_k}(s, Z)) - \sqrt{2\tau\eta}\,\xi\|^2\Big]. \tag{7}$$

This yields a principled and tractable implementation of Wasserstein proximal policy gradient within the implicit policy class, where actions are nudged by the critic's gradient and simultaneously diffused by Gaussian noise to encourage exploration. In practical implementation, we ignore the entropy term in $\nabla_a Q_\tau$, in a way similar to the approximation in SAC Haarnoja et al. (2018). For a rigorous treatment of Wasserstein information geometry, we refer readers to Chen & Li (2020); here, we offer a more intuitive argument. We also note that while Moskovitz et al. (2021) considers Wasserstein geometry, they solve the optimal coupling problem using kernel methods.

## 4 CONVERGENCE ANALYSIS

This section derives convergence guarantees for (WPPG), considering both exact (Section 4.1) and inexact (Section 4.2) Q-functions. Our analysis follows the roadmap in Lan (2021), but we provide special treatment for the Wasserstein metric in place of the Bregman divergence used therein.

### 4.1 EXACT Q-FUNCTION

We begin by stating the main assumptions. Let $\mathcal{P}_2(\mathbb{R}^d)$ denote the space of probability measures on $\mathbb{R}^d$ with finite second moments, equipped with the 2-Wasserstein metric. A probability distribution $\mu \in \mathcal{P}_2(\mathbb{R}^d)$ is said to satisfy $T_2$ transportation-information inequality, if for every $\nu \in \mathcal{P}_2(\mathbb{R}^d)$ it holds that

$$W_2^2(\nu, \mu) \leq \frac{2}{\lambda} \mathsf{KL}(\nu \| \mu). \qquad (T_2(\lambda))$$

**Assumption 1.** *For every state $s$, the followings hold:*

*(i) (Uniform $T_2(\lambda)$) There exists a constant $\lambda > 0$ such that $\pi(\cdot|s)$ satisfies $T_2(\lambda)$ for every $\pi(\cdot|s) \in \Pi(s)$.*

*(ii) (Boundedness) The reward is uniformly bounded and the action space $\mathcal{A}$ is bounded by $R$.*

*(iii) (Approximate realizability) There exists $\delta > 0$ such that for any iterate $\pi_k(\cdot|s)$ of (WPPG), there exists a $\pi_+(\cdot|s) \in \Pi(s)$ such that $\mathsf{KL}(\bar{\pi}(\cdot|s) \| \pi_+(\cdot|s)) \leq \delta^2$, where*

$$\bar{\pi}(\cdot \mid s) \in \operatorname{argmax}_{q(\cdot|s) \in \mathcal{P}(\mathcal{A})} \left\{ \langle Q^\pi(s, \cdot), q(\cdot|s) \rangle - \tau \mathsf{H}^q(s) - \frac{1}{2\eta_k} W_2^2\big(q(\cdot|s), \pi_k(\cdot \mid s)\big) \right\}. \qquad (8)$$

This assumption requires the policy class to be sufficiently expressive and regular. Consider, for example, a class of implicit policies defined in Section 3.1 with $g_\theta$ being a neural network and $\nu$ being a standard Gaussian distribution. Then, (i) can be ensured by designing the neural network to be Lipschitz continuous with respect to the latent variable inputs—this is because the standard Gaussian distribution satisfies $T_2(1)$ and Lipschitz transformations preserve $T_2$ inequalities (Ledoux, 2001). (iii) requires that the policy class to be sufficiently expressive to approximate the unconstrained Wasserstein proximal policy update (8) up to a small KL error.

**Remark 1.** *The bounded action space can be replaced by the condition that the policy $\pi(\cdot|s)$ has uniformly bounded second moment; see Lemma 4 and the comment afterwards in Appendix A.*

For a policy $\pi(\cdot|s)$ with initial state distribution $\rho$, we define

$$J_\rho(\pi) := \mathbb{E}^\pi_{s_0 \sim \rho}[V_\tau^\pi(s_0)].$$

Recall the definition of the soft value function (1), and by the soft Bellman optimality conditions, there exists an optimal policy $\pi^\star$ such that

$$V_\tau^{\pi^\star}(s) \geq V_\tau^\pi(s) \qquad \text{for all } s \in \mathcal{S}, \quad \text{and any } \pi.$$

Hence, optimizing $V^\pi(\cdot)$ state-wise is equivalent to optimizing any strictly positive weighted average of it. In particular, for any probability weights $\rho \in \mathcal{P}(\mathcal{S})$ with full support,

$$\pi^\star \in \operatorname{argmax}_\pi \mathbb{E}_{s \sim \rho}\left[V^\pi(s)\right] \quad \text{s.t.} \quad \pi(\cdot \mid s) \in \mathcal{P}(\mathcal{A}), \, \forall s \in \mathcal{S}.$$

While the initial distribution $\rho$ can be chosen arbitrarily, we follow Lan (2021) and set $\rho = \nu^*$—the stationary distribution induced by the optimal policy $\pi^*$—to ease the proof. Notably, our (WPPG) algorithm designed to optimize the objective (9) does not require access to $\nu^*$. We thereby define the objective function as

$$J(\pi) := J_{\nu^*}(\pi) = \mathbb{E}_{s \sim \nu^*}[V^\pi(s)], \qquad (9)$$

and our goal is to maximize $J_{\nu^*}(\pi)$ over all admissible policies:

$$\max_\pi J(\pi) \quad \text{s.t.} \quad \pi(\cdot \mid s) \in \mathcal{P}_\mathcal{A}, \, \forall s \in \mathcal{S}. \qquad (10)$$

Our main result in this subsection is as follows.

**Theorem 1** (Linear Convergence). *Suppose Assumption 1 holds and set the step size $\eta_k = \eta \geq \frac{1}{\gamma\lambda\tau}$. Then for any $k \geq 0$, the iterates of (WPPG) satisfy*

$$J(\pi_*) - J(\pi_k) + \lambda\tau\mathcal{D}(\pi_k, \pi^*) \leq \gamma^k \left[J(\pi^*) - J(\pi_0) + \lambda\tau\mathcal{D}(\pi_0, \pi^*)\right] + \mathcal{O}(\delta + \tau)$$

*where $J$ is defined in (9), and $\mathcal{D}(\pi_k, \pi^*) := \mathbb{E}_{s\sim\nu^*}\left[\frac{1}{2}\mathsf{W}_2^2\big(\pi_k(\cdot|s), \pi^*(\cdot|s)\big)\right]$.*

*Consequently, in order to achieve an error of $\mathcal{O}(\varepsilon + \delta + \tau)$, the required iteration complexity is*

$$\mathcal{O}\left(\frac{1}{1-\gamma}\,\log\frac{J(\pi^*) - J(\pi_0) + \lambda\tau\mathcal{D}(\pi_0, \pi^*)}{\varepsilon}\right).$$

This result shares some similarities with the linear convergence of mirror descent policy optimization presented in Lan (2021), but there are notable differences. First, our analysis is based on the Wasserstein distance, whereas theirs relies on the KL divergence, leading to distinct constants in the results. This discrepancy arises because the three-point lemma for KL/Bregman divergence does not apply in our setting. Instead, we leverage the geometry of the Wasserstein distance and the transportation-information inequality to establish new inequalities with different constants. From a methodological perspective, our work addresses the continuous action and state setting, explicitly accounting for the approximation error introduced by the implicit policy class. In contrast, their analysis is confined to finite states and finite actions, where explicit probability mass representations of policies enable exact solutions without the need to consider approximation error.

Another related result is Song et al. (2023, Theorem 5). In terms of the result, we focus on the entropy-regularized problem on continuous action spaces, whereas they consider the unregularized case on finite action spaces (with an extension to one-dimensional continuous spaces). The proof techniques are quite different. Their analysis builds on a bounded difference between Wasserstein policy update and policy-iteration using a uniform bound on the Wasserstein distance, but their analysis does not exploit any specific geometric property of the Wasserstein distance. As a result, their method requires an increasing step size schedule (corresponding to a decreasing Lagrangian multiplier $\beta$ in their paper), which implies that an $O(1/\varepsilon)$ step size is required to obtain an $\mathcal{O}(\varepsilon)$ solution. In contrast, in our method, if we set the policy realizability error $\delta = 0$ (matching their setting), we can achieve the same accuracy with a constant step size independent of $\varepsilon$.

### 4.2 INEXACT Q-FUNCTION

In practice, the exact action-value function $Q^{\pi_k}$ is rarely available, since computing it requires either full knowledge of the environment dynamics or an infinite number of Monte Carlo samples. Instead, one typically constructs a stochastic estimator $Q^{\pi_k,\xi_k}$ from finite trajectories, temporal-difference updates, or function approximation.

In this case, the (WPPG) update is defined by substituting the exact value function $Q^{\pi_k}$ in (29) with its stochastic estimator $Q^{\pi_k,\xi_k}$. Formally, the update rule is given by

$$\pi_{k+1}(\cdot|s) \in \mathrm{argmax}_{q(\cdot|s)\in\Pi(s)} \langle Q^{\pi_k,\xi_k}(s,\cdot), q\rangle - \tau H^\pi(s) - \frac{1}{2\eta_k}\mathsf{W}_2^2\big(q, \pi_k(\cdot|s)\big). \tag{11}$$

Such an estimator inevitably introduces both variance and bias, which can accumulate across iterations and significantly affect policy updates. To ensure that our analysis remains tractable while still capturing realistic scenarios, we impose mild conditions on the stochastic approximation and estimation error.

**Assumption 2.** *For each iteration $k \geq 0$, the stochastic estimator $Q^{\pi_k,\xi_k}$ satisfies*

$$\mathbb{E}_{\xi_k}\left[Q^{\pi_k,\xi_k}\right] = \bar{Q}^{\pi_k}, \tag{12}$$

$$\left\|\bar{Q}^{\pi_k} - Q^{\pi_k}\right\|_\infty \leq \epsilon_k, \tag{13}$$

$$\mathbb{E}_{\xi_k}\left[\left\|\nabla_a Q^{\pi_k,\xi_k} - \nabla_a Q^{\pi_k}\right\|_{2,\infty}^2\right] \leq \sigma_k^2. \tag{14}$$

*Where $\|\cdot\|_\infty$ is the uniform norm over $(s, a) \in \mathcal{S} \times \mathcal{A}$, and for any function $f$, $\|\cdot\|_{2,\infty}$ is defined as*

$$\|f\|_{2,\infty} := \sup_{(s,a)\in\mathcal{S}\times\mathcal{A}} \|f(s,a)\|_2.$$

This assumption is similar to Lan (2021), except that (14) concerns the action-gradient of the $Q$-function, which is needed in our Wasserstein policy update. We have the following convergence result.

**Theorem 2** (Linear Convergence). *Suppose Assumptions 1 and 2 hold, and for all $k \geq 0$, $\epsilon_k \leq \epsilon$, $\sigma_k \leq \sigma$ and $\|Q^{\pi_k, \xi_k}\|_\infty \leq B$. Then the iterates of (11) using step size $\eta_k = \eta \geq \frac{1}{\gamma \lambda \tau}$ satisfies*

$$\mathbb{E}_{\xi_{0:k-1}} \left[ J(\pi^*) - J(\pi_k) + \lambda \tau D(\pi_k, \pi^*) \right] \leq \gamma^k \left[ J(\pi^*) - J(\pi_0) + \lambda \tau D(\pi_0, \pi^*) \right] + \mathcal{O}(\delta + \tau + \epsilon + \sigma) \tag{15}$$

*where $J$ is defined in (9), and $\mathcal{D}(\pi_k, \pi^*) := \mathbb{E}_{s \sim \nu^*} \left[ \frac{1}{2} W_2^2 \big( \pi_k(\cdot|s), \pi^*(\cdot|s) \big) \right]$. Consequently, in order to achieve an error of $\mathcal{O}(\varepsilon + \tau + \delta + \epsilon + \sigma)$ in expectation, the required iteration complexity is*

$$\mathcal{O} \left( \frac{1}{1 - \gamma} \log \frac{J(\pi^*) - J(\pi_0) + \lambda \tau \mathcal{D}(\pi_0, \pi^*)}{\varepsilon} \right).$$

Compared with Theorem 1, this result involves additional bias arising from the estimation of the $Q$-function and variance from stochastic estimation of the $Q$-function. Nonetheless, our analysis shows that the total error does not accumulate across iterations, and the convergence guarantee only incurs an $\mathcal{O}(\delta + \tau + \epsilon + \sigma)$ term in the final bound, rather than growing with the number of iterations.

## 5 EXPERIMENTS

Our empirical study consists of two parts: comparative evaluation and ablation analysis. The comparative evaluation focuses on benchmarking our methods against representative baselines to assess overall performance. The ablation analysis investigates three key questions: (i) the effect of $\tau$ on WPPG, (ii) the impact of latent variable dimension on WPPG-I, and (iii) the role of double-$Q$ learning in WPPG, which is postponed to the appendix.

### 5.1 COMPARATIVE EVALUATION

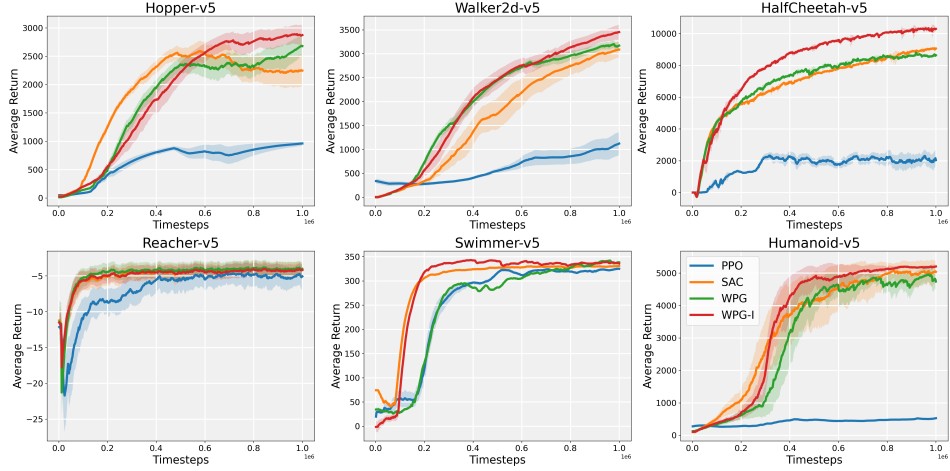

Figure 1: Training curves on MuJoCo continuous control benchmarks: Solid lines denote the mean episodic return, while shaded areas represent the 95% confidence interval computed over 10 independent evaluation runs with different random seeds.

**Evaluation Tasks** We evaluate our approach on a set of standard continuous control benchmarks from the MuJoCo suite [1], including Hopper-v5, Walker2d-v5, HalfCheetah-v5, Reacher-v5, Swimmer-v5, and Humanoid-v5. These tasks cover a wide range of difficulties: from relatively low-dimensional and easy-to-learn tasks such as Swimmer and Hopper, to high-dimensional and challenging tasks such as Humanoid.

**Baseline Models** We compare against three representative baselines: (i) PPO, a KL-proximal policy optimization method that employs clipped surrogate objectives to constrain successive policy updates and improve training stability (Schulman et al., 2017b); (ii) SAC, a stochastic actor–critic algorithm formulated as an entropy-regularized policy optimization method, which augments the reward with a maximum-entropy term to encourage exploration and improve stability (Haarnoja et al.,

---
[1]https://gymnasium.farama.org/environments/mujoco/

2018). (iii) WPO, a Wasserstein-proximal actor–critic algorithm that replaces the KL divergence commonly used in proximal methods with the Wasserstein distance, thereby constraining successive policy updates under the geometry of optimal transport. (Pfau et al., 2025). Our proposed methods include WPPG, with a Gaussian MLP policy actor, and WPPG-I, with an implicit MLP policy actor. Hyperparameters for PPO are taken from the RL Zoo project[2], while those for SAC and WPO follow Pfau et al. (2025).

**Experiment Setup** For fair comparison, WPPG and WPPG-I adopt the double-$Q$ technique, taking the minimum of two $Q$-functions for both critic targets and action gradients, while WPO follows the original single-$Q$ formulation. We also evaluate a single-$Q$ variant of WPPG for consistency, with its comparison to WPO revisited in the ablation study. To further ensure fairness, SAC is evaluated with entropy coefficient self-tuning disabled, since $\tau$ in WPPG and WPPG-I is also fixed rather than adaptively tuned while training. All off-policy methods share the same replay buffer structure. Additional implementation details for all methods and pseudo code of our algorithm are provided in the Appendix B.

**Results and Discussion** The learning curves for all tasks are shown in Figure 5. Across the six MuJoCo benchmarks, WPPG demonstrates performance comparable to SAC, which can be attributed to the fact that both methods employ Gaussian MLP policies. This observation suggests that Wasserstein geometry can match, and in some cases even surpass, the effectiveness of KL-based geometry in policy optimization. More importantly, WPPG-I consistently outperforms all baselines, achieving superior convergence speed and higher returns across nearly all tasks. The success of WPPG-I further indicates that WPPG can be naturally extended to implicit policy classes: although we use only a simple MLP-based implicit policy here, the framework readily accommodates richer architectures. We refer readers to the Appendix B.1 and B.2 for a detailed comparison between the two algorithms. In contrast, PPO lags behind due to slower learning and lower asymptotic performance, while WPO suffers from unstable convergence on challenging environments such as Humanoid and Swimmer and even fails to learn in Reacher. Overall, these results highlight that WPPG preserves the sample efficiency of off-policy actor-critic methods, while WPPG-I not only inherits these advantages but also demonstrates a consistent and significant margin over all baselines.

## 5.2 ABLATION STUDY

**Ablation on $\tau$.** In the preceding analysis, we showed that the parameter $\tau$ originates from entropy regularization of the policy. Unlike SAC and related methods that explicitly add an entropy penalty term into the $Q$-function fitting objective, WPPG does not require such a penalty. Instead, Gaussian noise is injected when computing the movement direction of action samples, where the scale of the injected noise $\tau$ corresponds to the magnitude of the entropy penalty.

To study the impact of $\tau$, we conducted ablation experiments on Humanoid environment. As illustrated in Figure 2, on Humanoid we observe that injecting noise with $\tau$ in the range $[0, 0.01]$ significantly accelerates convergence, while larger values $0.1$ slows it down. This reflects a clear exploration–exploitation trade-off: noise injection encourages the policy to maintain entropy, thereby enabling exploration of richer reward information, but excessive noise hampers the ability of $\nabla_a Q(s, a)$ to provide useful guidance for policy updates.

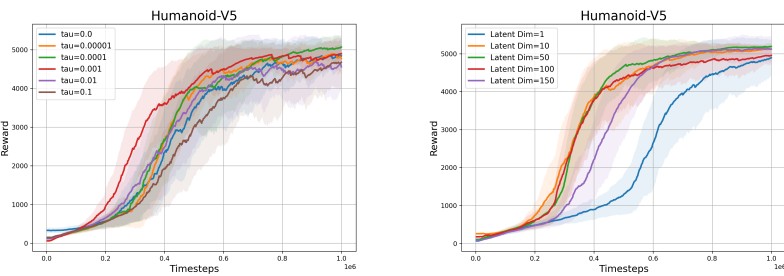

Figure 2: Ablation study on $\tau$ (left) and *Latent Dimension* (right).

---

[2] https://github.com/DLR-RM/rl-baselines3-zoo

**Ablation on Latent Dimension**    We further evaluate the effect of the dimension of latent variable $z$ in our implicit generative model on WPPG-I in the Humanoid environment. When the latent dimension is as small as 1, the model learns slowly due to insufficient stochasticity. With moderate dimensions (e.g., 10, 50, 100), learning is significantly accelerated, indicating that a reasonable amount of latent variables enhances exploration without overwhelming the policy. However, when the latent dimension becomes too large (e.g., 150), excessive non-informative variables begin to dominate the input and degrade learning speed. Empirically, we find that setting the latent dimension about one-third of the state dimension provides a good balance between exploration and stability.

**Ablation on Double Q Trick**    We also evaluated the single-$Q$ variant of WPPG across all environments, and found that it outperforms WPO on nearly every task, with the corresponding results provided in the Appendix B.4. In addition, consistent with prior findings, adopting double-$Q$ further improves WPPG by both stabilizing training and enhancing overall performance.

## 6    LIMITATIONS AND CONCLUSION

**Limitations**    Our empirical study focuses on standard continuous-control benchmarks, while the performance of WPPG and WPPG-I on ultra high-dimensional and more complex tasks remains unexplored. Moreover, the implicit policy formulation admits natural extensions to richer and more expressive classes, such as diffusion-based policies, which we have not yet investigated. Finally, whether Wasserstein geometry can serve as a foundation for reinforcement learning fine-tuning of large language models (LLMs) is an open and promising direction. We leave these broader applications to future work.

**Conclusion**    In this work, we proposed Wasserstein Proximal Policy Gradient (WPPG), a novel framework for policy optimization that leverages Wasserstein geometry to design proximal updates directly in distribution space. Our method eliminates the need for policy densities or score functions, making it naturally applicable to implicit policies. Theoretically, we established linear convergence guarantees under entropy regularization and a transport-entropy condition, covering both exact and approximate value function settings.To the best of our knowledge, this work is an early pioneering attempt to employ Wasserstein geometry for establishing global convergence guarantees. Empirically, WPPG demonstrates competitive performance against strong baselines, while its implicit extension WPPG-I consistently outperforms them across challenging continuous-control benchmarks. These results highlight the potential of Wasserstein geometry as a principled alternative to KL-based methods in reinforcement learning.

**Reproducibility Statement.**    Our entire implementation is built on top of the Stable Baselines3 (SB3) interface, ensuring compatibility and transparency. We provide detailed pseudocode for both WPPG and WPPG-I, together with comprehensive hyperparameter specifications, in the Appendix B.1. These materials cover nearly all technical details of our approach, and the pseudocode alone suffices to fully reproduce the proposed models. In addition, our experiments follow standardized training and evaluation protocols across tasks, which further supports reproducibility and comparability with prior baselines.

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

**LLM Usage Statement**   We used large language models (LLMs) only for language editing and polishing of the manuscript, and for occasional assistance in writing boilerplate code such as plotting scripts, LaTeX formatting, and Stable Baselines3 (SB3) wrapper templates. All research ideas, algorithmic designs, theoretical analyses, and experimental implementations are original and conducted by the authors.

## A   THEORETICAL DERIVATIONS AND PROOFS

We begin by showing a fundamental computation of the functional gradient of the value function $V^\pi_\tau(s_0)$ with respect to $\pi$.

**Lemma 1** (Log-derivative of the trajectory law). *Consider a zero-mass perturbation $\delta\pi(\cdot|s)$ (i.e. $\int_\mathcal{A} \delta\pi(da|s) = 0$ for all $s$) and the policy path $\pi_\epsilon = \pi + \epsilon\,\delta\pi$, with $|\epsilon|$ small so that $\pi_\epsilon \geq 0$ and $\mathrm{supp}(\delta\pi) \subseteq \{\pi > 0\}$. Let $\mathbb{P}_{\pi_\epsilon}$ be the path measure of the Markov chain induced by $\pi_\epsilon$ and the transition kernel $P(s'|s, a)$ from an initial law $\rho_0$. For any finite horizon $T$ and any integrable test functional $F$ of $\tau_{0:T} = (s_0, a_0, \ldots, s_T)$,*

$$\frac{d}{d\epsilon}\bigg|_{\epsilon=0} \int F(\tau_{0:T})\, dP_{\pi_\epsilon}(\tau_{0:T}) = \int F(\tau_{0:T}) \left( \sum_{u=0}^{T-1} \frac{\delta\pi(a_u \mid s_u)}{\pi(a_u \mid s_u)} \right) dP_\pi(\tau_{0:T}). \qquad (16)$$

*If $F$ is dominated by an integrable function uniformly, then letting $T \to \infty$ and applying dominated convergence yields*

$$\frac{d}{d\epsilon}\bigg|_{\epsilon=0} \int F(\tau)\, dP_{\pi_\epsilon}(\tau) = \int F(\tau) \left( \sum_{u=0}^{\infty} \frac{\delta\pi(a_u \mid s_u)}{\pi(a_u \mid s_u)} \right) dP_\pi(\tau). \qquad (17)$$

*Proof.* Write the truncated path density under $\pi_\epsilon$ as

$$d\mathbb{P}_{\pi_\epsilon}(\tau_{0:T}) = \rho_0(s_0) \prod_{t=0}^{T-1} \Big[ \pi_\epsilon(a_t \mid s_t)\, P(s_{t+1} \mid s_t, a_t) \Big].$$

Only $\pi_\epsilon$ depends on $\epsilon$, hence by the product rule,

$$\frac{d}{d\epsilon} \prod_{t=0}^{T-1} \pi_\epsilon(a_t \mid s_t) = \Big( \prod_{t=0}^{T-1} \pi_\epsilon(a_t \mid s_t) \Big) \sum_{u=0}^{T-1} \frac{\frac{d}{d\epsilon} \pi_\epsilon(a_u \mid s_u)}{\pi_\epsilon(a_u \mid s_u)}.$$

Evaluating at $\epsilon = 0$ gives

$$\frac{d}{d\epsilon} \pi_\epsilon(a_u \mid s_u)\bigg|_{\epsilon=0} = \delta\pi(a_u \mid s_u),$$

hence (16) follows by dominated convergence under the stated integrability. $\qquad \square$

**Lemma 2** (Functional policy gradient of Entropy). *Fix a state $s \in \mathcal{S}$. Let the policy at $s$ admit a density $\pi(\cdot|s)$ w.r.t. a reference measure $a$ (and write $h^\pi(s) := h(\pi(\cdot|s))$) with*

$$\mathsf{H}^\pi(s) \;=\; \int_\mathcal{A} \pi(a|s)\, \log \pi(a|s)\, a.$$

*For any zero-mass direction $\delta\pi(\cdot|s)$ (i.e. $\int_\mathcal{A} \delta\pi(a|s) = 0$) and the perturbation $\pi_\epsilon(\cdot|s) = \pi(\cdot|s) + \epsilon\,\delta\pi(\cdot|s)$, provided $\pi_\epsilon(\cdot|s)$ stays nonnegative for $|\epsilon|$ small and $\pi(a|s) > 0$ on its support, the Gâteaux derivative of $h^\pi(s)$ in the direction $\delta\pi(\cdot|s)$ is*

$$\frac{\delta\mathsf{H}^\pi(s)}{\delta\pi} := \frac{d}{d\epsilon}\bigg|_{\epsilon=0} \mathsf{H}^{\pi_\epsilon}(s) \;=\; \int_\mathcal{A} 1 + \log \pi(a|s)\, \delta\pi(da|s). \qquad (18)$$

*Proof.* Let $\pi_\epsilon(a) = \pi(a|s) + \epsilon\,\delta\pi(a|s)$. Then

$$\lim_{\epsilon\to 0} \frac{\mathsf{H}^{\pi_\epsilon}(s) - \mathsf{H}^\pi(s)}{\epsilon} = \lim_{\epsilon\to 0} \frac{1}{\epsilon}\Big( \int \pi_\epsilon(a|s)\, \log \pi_\epsilon(a|s)\, da - \int \pi(a|s)\, \log \pi(a|s)\, da \Big)$$

$$= \lim_{\epsilon\to 0} \frac{1}{\epsilon}\Big( \int (\pi(a|s) + \epsilon\,\delta\pi(a|s))\, \log(\pi(a|s) + \epsilon\,\delta\pi(a|s))\, da - \int \pi(a|s)\, \log \pi(a|s)\, da \Big)$$

$$= \int \lim_{\epsilon\to 0} \frac{\pi(a|s)\big(\ln(\pi(a|s) + \epsilon\,\delta\pi(a|s)) - \ln \pi(a|s)\big)}{\epsilon} + \delta\pi(a|s) \ln(\pi(a|s) + \epsilon\delta\pi(a|s))\, da$$

$$= \int (1 + \ln \pi)\, d\delta\pi$$

which implies that $\frac{\delta\mathsf{H}^\pi(s)}{\delta\pi}(a) = 1 + \ln \pi(a|s)$. $\qquad \square$

**Lemma 3** (Functional policy gradient under entropy regularization). *Then for any zero-mass direction $\delta\pi$, the Gâteaux derivative (functional gradient)*

$$\frac{\delta V_\tau^\pi(s_0)}{\delta \pi}(s,a) = \frac{1}{1-\gamma}\, d_{s_0}^\pi(s)\left(Q_\tau^\pi(s,a) - \tau(1 + \ln \pi(a|s))\right), \tag{19}$$

*Proof.* We should be very careful here: when computing the functional gradient of $V_\tau^\pi(s)$ with respect to $\pi$, there are two contributing parts - one from the MDP dynamics and one from the entropy regularization. Let $V_\tau^\pi(s) := \mathbb{E}_{\pi,s_0=s}\left[\sum_{t\geq 0}\gamma^t\left(r(s_t,a_t) - \tau\mathsf{H}^\pi(s_t)\right)\right]$ By Lemma 1 with $F(\tau) = \sum_{t\geq 0}\gamma^t\left(r(s_t,a_t) - \tau\mathsf{H}^\pi(s_t)\right)$,

$$\left.\frac{\mathrm{d}}{\mathrm{d}\epsilon}\right|_{\epsilon=0} V_\tau^{\pi_\epsilon} = \sum_{t\geq 0}\gamma^t\,\mathbb{E}_\pi\left[\left(r(s_t,a_t) - \tau\mathsf{H}^\pi(s_t)\right)\sum_{u=0}^{t}\frac{\delta\pi(a_u|s_u)}{\pi(a_u|s_u)}\right] - \sum_{t\geq 0}\gamma^t\,\mathbb{E}_\pi\left[\left.\frac{\mathrm{d}}{\mathrm{d}\epsilon}\right|_{\epsilon=0}\tau\mathsf{H}^{\pi_\epsilon}(s)\right]$$

*(I) Trajectory-law part.*

$$(\text{I}) = \sum_{u\geq 0}\mathbb{E}_\pi\left[\frac{\delta\pi(a_u|s_u)}{\pi(a_u|s_u)}\sum_{t\geq u}\gamma^t\left(r(s_t,a_t) - \tau\mathsf{H}^\pi(s_t)\right)\right] = \sum_{u\geq 0}\gamma^u\,\mathbb{E}_\pi\left[\frac{\delta\pi(a_u|s_u)}{\pi(a_u|s_u)}\,Q_\tau^\pi(s_u,a_u)\right].$$

Condition on $s_u$ and expand over $a$ to cancel $\pi$, then sum over time and use $\sum_{u\geq 0}\gamma^u\,\mathbb{P}_\pi(s_u = s \mid s_0) = \frac{1}{1-\gamma}d_{s_0}^\pi(s)$ to get

$$(\text{I}) = \iint_{\mathcal{S}\times\mathcal{A}}\frac{1}{1-\gamma}d_{s_0}^\pi(s)\,Q_\tau^\pi(s,a)\,\delta\pi(a|s)\mathrm{s}\mathrm{a}.$$

*(II) Explicit entropy part.* By Lemma 2 for each state $s$, $\mathsf{H}^\pi(s) = -\int \pi(a|s)\log\pi(a|s)\,\mathrm{a}$ , we have

$$\left.\frac{\mathrm{d}}{\mathrm{d}\epsilon}\right|_{\epsilon=0}\mathsf{H}^{\pi_\epsilon}(s) = \int_{\mathcal{A}}[1 + \log\pi(a|s)]\,\delta\pi(a|s)\mathrm{d}a,$$

hence

$$(\text{II}) = \sum_{t\geq 0}\gamma^t\,\mathbb{E}_\pi[\tau\left.\frac{\mathrm{d}}{\mathrm{d}\epsilon}\right|_{\epsilon=0}\mathsf{H}^{\pi_\epsilon}(s)] = \iint_{\mathcal{S}\times\mathcal{A}}\frac{1}{1-\gamma}d_{s_0}^\pi(s)\,\tau[1 + \log\pi(a|s)]\,\delta\pi(a|s)\mathrm{d}a\mathrm{d}s.$$

Finally, we have

$$\frac{\delta V_\tau^\pi(s_0)}{\delta\pi}(s,a) = \frac{1}{1-\gamma}d_{s_0}^\pi(s)\left(Q_\tau^\pi(s,a) - \tau(1 + \ln\pi(a|s))\right)$$

$\square$

**Lemma 4** (Entropy upper bound from second moment). *Let $X \in \mathbb{R}^d$ be absolutely continuous with differential entropy $h(X)$. Assume*

$$\mathbb{E}\|X\|^2 \leq R^2.$$

*Then*

$$h(X) \leq \frac{d}{2}\log\left(2\pi e\,\frac{R^2}{d}\right).$$

.

*Proof.* Let $\mu = \mathbb{E}X$ and $\Sigma = \mathrm{Cov}(X)$. Among all distributions with covariance $\Sigma$, the Gaussian maximizes differential entropy, hence

$$h(X) \leq \frac{1}{2}\log\left((2\pi e)^d\det\Sigma\right).$$

Also,

$$\mathrm{tr}(\Sigma) = \mathbb{E}\|X - \mu\|^2 \leq \mathbb{E}\|X\|^2 \leq R^2.$$

Let $\lambda_1,\ldots,\lambda_d$ be eigenvalues of $\Sigma$. Then $\sum_i \lambda_i = \mathrm{tr}(\Sigma) \leq R^2$ and $\det\Sigma = \prod_i \lambda_i$. By AM–GM,

$$\det\Sigma \leq \left(\frac{1}{d}\sum_{i=1}^{d}\lambda_i\right)^d \leq \left(\frac{R^2}{d}\right)^d.$$

Substituting gives

$$h(X) \leq \frac{1}{2}\log\left((2\pi e)^d\left(\frac{R^2}{d}\right)^d\right) = \frac{d}{2}\log\left(2\pi e\,\frac{R^2}{d}\right).$$

.

$\square$

With this lemma, if the second moments of policies are uniformly upper bounded, then their entropies are also uniformly upper bounded, so are the soft-Q functions $Q_\tau^\pi$ defined in (1).

Then we present the performance differance lemma which is the most important thing in reinforcement learning.

**Lemma 5** (Entropy Regularized Performance Difference Lemma). *(Lan, 2023, Lemma2) For any two feasible policies $\pi$ and $\pi'$, we have*

$$V_\tau^{\pi'}(s) - V_\tau^\pi(s) = \frac{1}{1-\gamma} \mathbb{E}_{s' \sim d_s^{\pi'}} \left[ \langle A_\tau^\pi(s', \cdot), \pi'(\cdot|s') \rangle - \tau \mathsf{H}^{\pi'}(s') + \tau \mathsf{H}^\pi(s') \right],$$

*where $A_\tau^\pi(s', a) := Q_\tau^\pi(s', a) - V_\tau^\pi(s')$.*

For completeness, we provide the proof here and adapt it to our notation.

*Proof.* For simplicity, let us denote $\xi^{\pi'}(s_0)$ the random process $(s_t, a_t, s_{t+1})$, $t \geq 0$, generated by following the policy $\pi'$ starting with the initial state $s_0$. It then follows from the definition of $V_\tau^{\pi'}$ that

$$V_\tau^{\pi'}(s) - V_\tau^\pi(s)$$

$$= \mathbb{E}_{\xi^{\pi'}(s)} \left[ \sum_{t=0}^\infty \gamma^t \left( r(s_t, a_t) - \tau \mathsf{H}^{\pi'}(s_t) \right) \right] - V_\tau^\pi(s)$$

$$= \mathbb{E}_{\xi^{\pi'}(s)} \left[ \sum_{t=0}^\infty \gamma^t \left( r(s_t, a_t) - \tau \mathsf{H}^{\pi'}(s_t) + V_\tau^\pi(s_t) - V_\tau^\pi(s_t) \right) \right] - V_\tau^\pi(s)$$

$$\overset{(1)}{=} \mathbb{E}_{\xi^{\pi'}(s)} \left[ \sum_{t=0}^\infty \gamma^t \left( r(s_t, a_t) - \tau \mathsf{H}^{\pi'}(s_t) + \gamma V_\tau^\pi(s_{t+1}) - V_\tau^\pi(s_t) \right) \right]$$

$$\quad + \mathbb{E}_{\xi^{\pi'}(s)} \left[ V_\tau^\pi(s_0) \right] - V_\tau^\pi(s)$$

$$\overset{(2)}{=} \mathbb{E}_{\xi^{\pi'}(s)} \left[ \sum_{t=0}^\infty \gamma^t \left( r(s_t, a_t) - \tau \mathsf{H}^{\pi'}(s_t) + \gamma V_\tau^\pi(s_{t+1}) - V_\tau^\pi(s_t) \right) \right]$$

$$= \mathbb{E}_{\xi^{\pi'}(s)} \left[ \sum_{t=0}^\infty \gamma^t \left( r(s_t, a_t) - \tau \mathsf{H}^\pi(s_t) + \gamma V_\tau^\pi(s_{t+1}) - V_\tau^\pi(s_t) - \tau \mathsf{H}^{\pi'}(s_t) + \tau \mathsf{H}^\pi(s_t) \right) \right]$$

$$\overset{(3)}{=} \mathbb{E}_{\xi^{\pi'}(s)} \left[ \sum_{t=0}^\infty \gamma^t \left( Q_\tau^\pi(s_t, a_t) - V_\tau^\pi(s_t) - \tau \mathsf{H}^{\pi'}(s_t) + \tau \mathsf{H}^\pi(s_t) \right) \right].$$

where (1) follows by taking the term $V_\tau^\pi(s_0)$ outside the summation, (2) follows from the fact that $\mathbb{E}_{\xi^{\pi'}(s)}[V_\tau^\pi(s_0)] = V_\tau^\pi(s)$ since the random process starts with $s_0 = s$, and (3) follows from 1. The previous conclusion then imply that

$$V_\tau^{\pi'}(s) - V_\tau^\pi(s)$$

$$= \frac{1}{1-\gamma} \sum_{s' \in \mathcal{S}} \sum_{a' \in \mathcal{A}} d_\gamma^{\pi'}(s') \pi'(a'|s') \left[ A_\tau^\pi(s', a') + \tau \mathsf{H}^{\pi'}(s') - \tau \mathsf{H}^\pi(s') \right]$$

$$= \frac{1}{1-\gamma} \sum_{s' \in \mathcal{S}} d_\gamma^{\pi'}(s') \left[ A_\tau^\pi(s', \pi'(\cdot|s')) - \tau \mathsf{H}^{\pi'}(s') + \tau \mathsf{H}^\pi(s') \right],$$

which immediately implies the result. $\square$

Remember our defination of $\nu^*$ as the steady state distribution induced by $\pi^*$.

**Lemma 6.** *(Lan, 2023, Lemma3)*

$$\mathbb{E}_{s \sim \nu^*} \left[ Q_\tau^\pi(s, \cdot), \pi^*(\cdot|s) - \pi(\cdot|s) - \tau \mathsf{H}^{\pi^*}(s) + \tau \mathsf{H}^\pi(s) \right] = \mathbb{E}_{s \sim \nu^*} \left[ (1-\gamma) \left( V_\tau^{\pi^*}(s) - V_\tau^\pi(s) \right) \right]. \tag{20}$$

For completeness, we provide the proof here and adapt it to our notation.

*Proof.* It follows from Lemma 5 (with $\pi' = \pi^*$) that

$$(1-\gamma) \left[ V_\tau^{\pi^*}(s) - V_\tau^\pi(s) \right] = \mathbb{E}_{s' \sim d_s^{\pi^*}} \left[ A_\tau^\pi(s', \cdot), \pi^*(\cdot|s') + \tau \mathsf{H}^\pi(s') - \tau \mathsf{H}^{\pi^*}(s') \right].$$

Noting that:

$$\langle A_\tau^\pi(s', \cdot), \pi^*(\cdot|s') \rangle = \langle Q_\tau^\pi(s', \cdot), \pi^*(\cdot|s') \rangle - V_\tau^\pi(s') \tag{21}$$

$$= \langle Q_\tau^\pi(s', \cdot), \pi^*(\cdot|s') \rangle - \langle Q_\tau^\pi(s', \cdot), \pi(\cdot|s') \rangle \tag{22}$$

$$= \langle Q_\tau^\pi(s', \cdot), \pi^*(\cdot|s') - \pi(\cdot|s') \rangle, \tag{23}$$

Combining the above two relations and taking expectation w.r.t. $\nu^*$, we obtain

$$(1-\gamma)\mathbb{E}_{s\sim\nu^*}\left[V_\tau^{\pi^*}(s) - V_\tau^\pi(s)\right] = \mathbb{E}_{s\sim\nu^*,\, s'\sim d_s^{\pi^*}}\left[\langle Q_\tau^\pi(s', \cdot), \pi^*(\cdot|s') - \pi(\cdot|s') \rangle + \tau\mathsf{H}^\pi(s') - \tau\mathsf{H}^{\pi^*}(s')\right]$$

$$= \mathbb{E}_{s\sim\nu^*}\left[Q_\tau^\pi(s, \cdot), \pi^*(\cdot|s) - \pi(\cdot|s) - \tau\mathsf{H}^{\pi^*}(s)\right] + \tau\mathsf{H}^\pi(s)$$

where the second identity is due to $\nu^*$ is the steady state distribution induced by $\pi^*$. $\qquad\square$

Next we will show a geometry property of the squared $\mathsf{W}_2$ distance that we repeatedly leverage in our convergence analysis.

**Lemma 7.** *Let $\nu \in \mathcal{P}_2(\mathbb{R}^d)$. For any $\rho, \mu \in \mathcal{P}_2(\mathbb{R}^d)$, let $(\varphi^{\rho\to\nu}, \psi^{\rho\to\nu})$ be an optimal Kantorovich dual pair for the cost $c(x,y) = \frac{1}{2}\|x-y\|^2$ between $(\rho, \nu)$, i.e.*

$$\varphi(x) + \psi(y) \leq \frac{1}{2}\|x-y\|^2 \quad \text{for all } x, y \in \mathbb{R}^d \tag{24}$$

$$\int \varphi^{\rho\to\nu} d\rho + \int \psi^{\rho\to\nu} d\nu = \frac{1}{2}\mathsf{W}_2^2(\rho, \nu). \tag{25}$$

*Then, for every $\mu \in \mathcal{P}_2(\mathbb{R}^d)$,*

$$\frac{1}{2}\mathsf{W}_2^2(\mu, \nu) \geq \frac{1}{2}\mathsf{W}_2^2(\rho, \nu) + \int_{\mathbb{R}^d} \varphi^{\rho\to\nu}(x)\big(\mu - \rho\big)(dx). \tag{26}$$

*Proof.* Recall the dual formulation (valid for any $\mu \in \mathcal{P}_2(\mathbb{R}^d)$):

$$\frac{1}{2}\mathsf{W}_2^2(\mu, \nu) = \sup_{\varphi, \psi}\left\{ \int \varphi\, d\mu + \int \psi\, d\nu : \varphi(x) + \psi(y) \leq \frac{1}{2}\|x-y\|^2 \right\}. \tag{27}$$

Since the feasibility constraint (24) is pointwise in $(x,y)$, the optimal pair $(\varphi^{\rho\to\nu}, \psi^{\rho\to\nu})$ for $(\rho, \nu)$ is also a feasible pair in the supremum (27) for $(\mu, \nu)$. Therefore,

$$\frac{1}{2}\mathsf{W}_2^2(\mu, \nu) \geq \int \varphi^{\rho\to\nu} d\mu + \int \psi^{\rho\to\nu} d\nu. \tag{28}$$

By optimality for $(\rho, \nu)$, we have (25). Subtracting $\int \varphi^{\rho\to\nu} d\rho$ on the right-hand side of (28) yields

$$\frac{1}{2}\mathsf{W}_2^2(\mu, \nu) \geq \left( \int \varphi^{\rho\to\nu} d\rho + \int \psi^{\rho\to\nu} d\nu \right) + \int \varphi^{\rho\to\nu} d(\mu-\rho) = \frac{1}{2}\mathsf{W}_2^2(\rho, \nu) + \int \varphi^{\rho\to\nu} d(\mu-\rho),$$

which is exactly (26). $\qquad\square$

**Lemma 8** (Wasserstein proximal one step inequality). *Fix a state $s \in \mathcal{S}$ and let the per-state JKO/proximal update be*

$$\bar{\pi}_{k+1}(\cdot|s) \in \operatorname{argmax}_{q\in\mathcal{P}_\mathcal{A}}\left\{ \langle Q^{\pi_k}(s, \cdot), q \rangle - \tau\mathsf{H}^q(s) - \frac{1}{2\eta_k}\mathsf{W}_2^2(q, \pi_k(\cdot|s)) \right\}, \tag{29}$$

*where $\langle f, q \rangle := \int_{a\in\mathcal{A}} f(a)\, q(a) da$. Then for any competitor $p \in \Delta_\mathcal{A}$,*

$$\eta_k\Big( \langle Q^{\pi_k}(s, \cdot), p - \bar{\pi}_{k+1}(\cdot|s) \rangle - \tau\mathsf{H}^p(s) + \tau\mathsf{H}^{\bar{\pi}_{k+1}}(s) \Big) + \tfrac{1}{2}\mathsf{W}_2^2\big(\bar{\pi}_{k+1}(\cdot|s), \pi_k(\cdot|s)\big)$$

$$\leq \tfrac{1}{2}\mathsf{W}_2^2\big(p, \pi_k(\cdot|s)\big) - \tfrac{\eta_k\lambda\tau}{2}\mathsf{W}_2^2\big(p, \bar{\pi}_{k+1}(\cdot|s)\big).$$

*Proof.* Let $\varphi^{\bar{\pi}_{k+1}\to\pi_k}(s, \cdot)$ be a Kantorovich potential for the pair $\big(\bar{\pi}_{k+1}(\cdot|s), \pi_k(\cdot|s)\big)$ under cost $c(a, a') = \frac{1}{2}\|a - a'\|^2$. Note that $\langle Q^{\pi_k}(s, \cdot), q \rangle$ is a linear functional of $q$, $-\tau\mathsf{H}^q(s)$ is strongly concave in $q$, and $-\frac{1}{2\eta_k}\mathsf{W}_2^2(q, \pi_k(\cdot|s))$ is concave in $q$. The first-order optimality condition of the concave program (29) states that

$$\Big\langle \eta_k\big( Q^{\pi_k}(s, \cdot) - \tau(1+\ln\bar{\pi}_{k+1}(\cdot|s)) \big) - \varphi^{\bar{\pi}_{k+1}\to\pi_k}(\cdot, a),\ p(\cdot|s) - \bar{\pi}_{k+1}(\cdot|s) \Big\rangle \leq 0, \qquad \forall p(\cdot|s) \in \mathcal{P}_\mathcal{A}. \tag{30}$$

Next, apply 7 for $F(p) = \frac{1}{2}\mathsf{W}_2^2(p, \pi_k(\cdot|s))$ and arbitrary $p(\cdot|s)$:

$$\tfrac{1}{2}\mathsf{W}_2^2\big(p(\cdot|s), \pi_k(\cdot|s)\big) \geq \tfrac{1}{2}\mathsf{W}_2^2\big(\bar{\pi}_{k+1}(\cdot|s), \pi_k(\cdot|s)\big) + \big\langle \varphi^{\bar{\pi}_{k+1}\to\pi_k}(s, \cdot),\ p(\cdot|s) - \bar{\pi}_{k+1}(\cdot|s) \big\rangle. \tag{31}$$

Rearranging (31) gives

$$\big\langle \varphi^{\bar{\pi}_{k+1}\to\pi_k}, p - \bar{\pi}_{k+1} \big\rangle \leq \tfrac{1}{2}\mathsf{W}_2^2(p, \pi_k) - \tfrac{1}{2}\mathsf{W}_2^2(\bar{\pi}_{k+1}, \pi_k),$$

where the arguments $(\cdot|s)$ are omitted for readability. Plug this bound into optimal condition to obtain

$$\left\langle \eta_k \left( Q^{\pi_k}(s,\cdot) - \tau(1 + \ln \bar{\pi}_{k+1}) \right),\; p - \bar{\pi}_{k+1} \right\rangle + \tfrac{1}{2} W_2^2(\bar{\pi}_{k+1}, \pi_k) \leq \tfrac{1}{2} W_2^2(p, \pi_k), \qquad \forall\, p(\cdot|s) \in \mathcal{P}_{\mathcal{A}}. \tag{32}$$

By noting the two facts that

$$\langle 1, p - \bar{\pi}_{k+1}(\cdot|s) \rangle = 0,$$

and

$$\langle \ln \bar{\pi}_{k+1}(\cdot|s), p \rangle = \langle \ln \bar{\pi}_{k+1}(\cdot|s), p \rangle - \langle \ln p, p \rangle + \langle \ln p, p \rangle \tag{33}$$
$$= -\mathsf{KL}(p \| \bar{\pi}_{k+1}(\cdot|s)) + \mathsf{H}^p(s), \tag{34}$$

we have

$$\eta_k \left( \langle Q^{\pi_k}(s,\cdot), p - \bar{\pi}_{k+1}(\cdot|s) \rangle - \tau \mathsf{H}^p(s) + \tau \mathsf{H}^{\bar{\pi}_{k+1}}(s) \right) + \tfrac{1}{2} W_2^2\left( \bar{\pi}_{k+1}(\cdot|s), \pi_k(\cdot|s) \right)$$
$$\leq \tfrac{1}{2} W_2^2\left( p, \pi_k(\cdot|s) \right) - \tfrac{\eta_k \lambda \tau}{2} W_2^2\left( p, \bar{\pi}_{k+1}(\cdot|s) \right). \tag{35}$$

$\square$

**Lemma 9.** *For any $s \in \mathcal{S}$, we have*
$$V_\tau^{\bar{\pi}_{k+1}}(s) - V_\tau^{\pi_k}(s) \geq \langle Q_\tau^{\pi_k}(s,\cdot), \bar{\pi}_{k+1}(\cdot|s) - \pi_k(\cdot|s) \rangle - \tau \mathsf{H}^{\bar{\pi}_{k+1}}(s) + \tau \mathsf{H}^{\pi_k}(s). \tag{36}$$

*Proof.* It follows from Lemma 5 (with $\pi' = \pi^{k+1}$, $\pi = \pi^k$) that
$$V_\tau^{\bar{\pi}_{k+1}}(s) - V_\tau^{\pi_k}(s) = \frac{1}{1-\gamma} \mathbb{E}_{s' \sim d_s^{\pi_k}} \left[ \langle A_\tau^{\pi_k}(s',\cdot), \bar{\pi}_{k+1}(\cdot|s') \rangle - \tau \mathsf{H}^{\bar{\pi}_{k+1}}(s') + \tau \mathsf{H}^{\pi_k}(s') \right]. \tag{37}$$

And
$$\langle A_\tau^{\pi_k}(s',\cdot), \bar{\pi}_{k+1}(\cdot|s') \rangle = \langle Q_\tau^{\pi_k}(s',\cdot), \bar{\pi}_{k+1}(\cdot|s') \rangle - V_\tau^{\pi_k}(s')$$
$$= \langle Q_\tau^{\pi_k}(s',\cdot), \bar{\pi}_{k+1}(\cdot|s') \rangle - \langle Q_\tau^{\pi_k}(s',\cdot), \pi_k(\cdot|s') \rangle$$
$$= \langle Q_\tau^{\pi_k}(s',\cdot), \bar{\pi}_{k+1}(\cdot|s') - \pi_k(\cdot|s') \rangle.$$

Combining the two identities above, we obtain
$$V_\tau^{\bar{\pi}_{k+1}}(s) - V_\tau^{\pi_k}(s) = \frac{1}{1-\gamma} \mathbb{E}_{s' \sim d_s^{\bar{\pi}_{k+1}}} \left[ \langle Q_\tau^{\pi_k}(s',\cdot), \bar{\pi}_{k+1}(\cdot|s') - \pi_k(\cdot|s') \rangle - \tau \mathsf{H}^{\bar{\pi}_{k+1}}(s') + \tau \mathsf{H}^{\pi_k}(s') \right]. \tag{38}$$

Now we conclude from Lemma 8 with $p = \pi_k(\cdot|s')$ for any $s'$ that
$$\langle Q_\tau^{\pi_k}(s',\cdot), \bar{\pi}_{k+1}(\cdot|s') - \pi_k(\cdot|s') \rangle - \tau \mathsf{H}^{\bar{\pi}_{k+1}}(s') + \tau \mathsf{H}^{\pi_k}(s') \;\geq\; \frac{\eta_k \lambda \tau}{2} W_2^2\left( \pi_k(\cdot|s'), \bar{\pi}_{k+1}(\cdot|s') \right). \tag{39}$$

The previous two conclusions then clearly imply the result in (36).
It also follows from (39) that
$$\mathbb{E}_{s' \sim d_s^{\pi_k}} \left[ \langle Q_\tau^{\pi_k}(s',\cdot), \bar{\pi}_{k+1}(\cdot|s') - \pi_k(\cdot|s') \rangle - \tau \mathsf{H}^{\bar{\pi}_{k+1}}(s') + \tau \mathsf{H}^{\pi_k}(s') \right]$$
$$\geq\; d_s^{\pi_k}(s) \left[ \langle Q_\tau^{\pi_k}(s,\cdot), \bar{\pi}_{k+1}(\cdot|s) - \pi_k(\cdot|s) \rangle - \tau \mathsf{H}^{\bar{\pi}_{k+1}}(s) + \tau \mathsf{H}^{\pi_k}(s) \right]$$
$$\geq\; (1-\gamma) \left[ \langle Q_\tau^{\pi_k}(s,\cdot), \bar{\pi}_{k+1}(\cdot|s) - \pi_k(\cdot|s) \rangle - \tau \mathsf{H}^{\bar{\pi}_{k+1}}(s) + \tau \mathsf{H}^{\pi_k}(s) \right],$$

where the last inequality follows from the fact that $d_s^{\pi_k}(s) \geq (1-\gamma)$ due to the definition of $d_s^{\pi_k}$ and $s_0 = s$ with probability one. Then by (38) and the above inequality, the claim follows. $\square$

**Lemma 10** (Function Approximation Error). *Under Assumption 1, we can bound the error of the value function induced by the function approximation step, i.e., for any $s$,*
$$|V_\tau^{\bar{\pi}_{k+1}}(s) - V_\tau^{\pi_{k+1}}(s)| \leq \mathcal{O}(\delta + \tau). \tag{40}$$

*Proof.* It follows from Lemma 5 (with $\pi' = \bar{\pi}_{k+1}$, $\pi = \pi_{k+1}$) that
$$V_\tau^{\bar{\pi}_{k+1}}(s) - V_\tau^{\pi_{k+1}}(s) = \frac{1}{1-\gamma} \mathbb{E}_{s' \sim d_s^{\bar{\pi}_{k+1}}} \left[ \langle Q_\tau^{\pi_{k+1}}(s',\cdot), \bar{\pi}_{k+1}(\cdot|s') - \pi_{k+1} \rangle - \tau \mathsf{H}^{\bar{\pi}_{k+1}}(s') + \tau \mathsf{H}^{\pi_{k+1}}(s') \right]. \tag{41}$$

For any $s'$, the first term is bounded by
$$\langle Q_\tau^{\pi_{k+1}}(s',\cdot), \bar{\pi}_{k+1}(\cdot|s') - \pi_{k+1} \rangle \leq \|Q_\tau^{\pi_{k+1}}(s',\cdot)\|_\infty \,\mathrm{TV}(\bar{\pi}_{k+1}(\cdot|s'), \pi_{k+1}(\cdot|s'))$$
$$\leq B\delta, \tag{42}$$

where the first inequality is due to Hölder's inequality, and the second follows from Pinsker's inequality, which says

$$\text{TV}(\bar{\pi}_{k+1}(\cdot|s'), \pi_{k+1}(\cdot|s')) \leq \sqrt{\tfrac{\text{KL}(\bar{\pi}_{k+1}(\cdot|s')\|\pi_{k+1}(\cdot|s'))}{2}} \leq \tfrac{\delta}{\sqrt{2}}.$$

By Lemma4, we have $\mathsf{H}^{\pi_{k+1}}(s') \leq \tfrac{d}{2}\log\left(2\pi e\,\tfrac{R^2}{d}\right)$, $\mathsf{H}^{\bar{\pi}_{k+1}}(s') \leq \tfrac{d}{2}\log\left(2\pi e\,\tfrac{R^2}{d}\right)$

For ease of notation, let $C := d\log\left(2\pi e\,\tfrac{R^2}{d}\right)$

Finally, we obtain

$$|V_\tau^{\bar{\pi}_{k+1}}(s) - V_\tau^{\pi_{k+1}}(s)| = \left|\frac{1}{1-\gamma}\mathbb{E}_{s'\sim d_s^{\bar{\pi}_{k+1}}}\left[\langle Q_\tau^{\pi_{k+1}}(s',\cdot), \bar{\pi}_{k+1}(\cdot|s') - \pi_{k+1}\rangle - \tau\mathsf{H}^{\bar{\pi}_{k+1}}(s') + \tau\mathsf{H}^{\pi_{k+1}}(s')\right]\right|$$

$$\leq \frac{1}{1-\gamma}\mathbb{E}_{s'\sim d_s^{\bar{\pi}_{k+1}}}\left[|\langle Q_\tau^{\pi_{k+1}}(s',\cdot), \bar{\pi}_{k+1}(\cdot|s') - \pi_{k+1}\rangle| + |\tau\mathsf{H}^{\bar{\pi}_{k+1}}(s') - \tau\mathsf{H}^{\pi_{k+1}}(s')|\right]$$

$$\leq \frac{B}{1-\gamma}\delta + \frac{C}{1-\gamma}\tau.$$

$$(43)$$

$\square$

**Theorem 1** (Linear Convergence). *Suppose Assumption 1 holds and set the step size $\eta_k = \eta \geq \tfrac{1}{\gamma\lambda\tau}$. Then for any $k \geq 0$, the iterates of (*WPPG*) satisfy*

$$J(\pi_*) - J(\pi_k) + \lambda\tau\mathcal{D}(\pi_k, \pi^*) \leq \gamma^k\left[J(\pi^*) - J(\pi_0) + \lambda\tau\mathcal{D}(\pi_0, \pi^*)\right] + \mathcal{O}(\delta + \tau)$$

*where $J$ is defined in (9), and $\mathcal{D}(\pi_k, \pi^*) := \mathbb{E}_{s\sim\nu^*}\left[\tfrac{1}{2}\,\mathsf{W}_2^2\big(\pi_k(\cdot|s), \pi^*(\cdot|s)\big)\right].$*

*Consequently, in order to achieve an error of $\mathcal{O}(\varepsilon + \delta + \tau)$, the required iteration complexity is*

$$\mathcal{O}\left(\frac{1}{1-\gamma}\,\log\frac{J(\pi^*) - J(\pi_0) + \lambda\tau\mathcal{D}(\pi_0, \pi^*)}{\varepsilon}\right).$$

*Proof.* By Lemma 8 with $p = \pi^*$, we have

$$\langle Q_\tau^{\pi_k}(s,\cdot), \pi^*(\cdot|s) - \pi_{k+1}(\cdot|s)\rangle - \tau\mathsf{H}^{\pi_*}(s) + \tau\mathsf{H}^{\pi_{k+1}}(s) + \frac{1}{2\eta_k}\mathsf{W}_2^2\big(\bar{\pi}_{k+1}(\cdot|s), \pi_k(\cdot|s)\big)$$

$$\leq \frac{1}{2\eta_k}\mathsf{W}_2^2\big(\pi_k(\cdot|s), \pi^*(\cdot|s)\big) - \frac{\lambda\tau}{2\eta_k}\mathsf{W}_2^2\big(\bar{\pi}_{k+1}(\cdot|s), \pi^*(\cdot|s)\big).$$

Combining with (36), we obtain

$$\left[\langle Q_\tau^{\pi_k}(s,\cdot), \pi^*(\cdot|s) - \pi_k(\cdot|s)\rangle - \tau\mathsf{H}^{\pi^*}(s) + \tau\mathsf{H}^{\pi_k}(s)\right]$$

$$+ \left[V_\tau^{\pi_k}(s) - V_\tau^{\bar{\pi}_{k+1}}(s)\right]$$

$$+ \frac{1}{2\eta_k}\mathsf{W}_2^2\big(\pi_k(\cdot|s), \bar{\pi}_{k+1}(\cdot|s)\big)$$

$$\leq \left[\langle Q_\tau^{\pi_k}(s,\cdot), \pi^*(\cdot|s) - \pi_k(\cdot|s)\rangle - \tau\mathsf{H}^{\pi^*}(s) + \tau\mathsf{H}^{\pi_k}(s)\right]$$

$$- \left[\langle Q_\tau^{\pi_k}(s,\cdot), \bar{\pi}_{k+1}(\cdot|s) - \pi_k(\cdot|s)\rangle - \tau\mathsf{H}^{\bar{\pi}_{k+1}}(s) + \tau\mathsf{H}^{\pi_k}(s)\right]$$

$$+ \frac{1}{2\eta_k}\mathsf{W}_2^2\big(\pi_k(\cdot|s), \bar{\pi}_{k+1}(\cdot|s)\big)$$

$$= \left(\langle Q_\tau^{\pi_k}(s,\cdot), \pi^*(\cdot|s) - \bar{\pi}_{k+1}(\cdot|s)\rangle - \tau\mathsf{H}^{\pi^*}(s) + \tau\mathsf{H}^{\bar{\pi}_{k+1}}(s)\right)$$

$$+ \frac{1}{2\eta_k}\mathsf{W}_2^2\big(\bar{\pi}_{k+1}(\cdot|s), \pi_k(\cdot|s)\big)$$

$$\leq \frac{1}{2\eta_k}\mathsf{W}_2^2\big(\pi_k(\cdot|s), \pi^*(\cdot|s)\big) - \frac{\lambda\tau}{2}\mathsf{W}_2^2\big(\bar{\pi}_{k+1}(\cdot|s), \pi^*(\cdot|s)\big).$$

Taking expectation with respect to $\nu^*$ on both sides of the inequality, we obtain

$$\mathbb{E}_{s\sim\nu^*}\Big[(1-\gamma)\big(V_\tau^{\pi^*}(s) - V_\tau^{\pi_k}(s)\big)\Big] + \mathbb{E}_{s\sim\nu^*}\Big[V_\tau^{\pi_k}(s) - V_\tau^{\bar{\pi}_{k+1}}(s)\Big]$$

$$+ \mathbb{E}_{s\sim\nu^*}\Big[\tfrac{1}{2\eta_k}\mathsf{W}_2^2\big(\pi_k(\cdot|s), \bar{\pi}_{k+1}(\cdot|s)\big)\Big]$$

$$\leq \mathbb{E}_{s\sim\nu^*}\Big[\tfrac{1}{2\eta_k}\mathsf{W}_2^2\big(\pi_k(\cdot|s), \pi^*(\cdot|s)\big) - \tfrac{\lambda\tau}{2}\mathsf{W}_2^2\big(\bar{\pi}_{k+1}(\cdot|s), \pi^*(\cdot|s)\big)\Big].$$

Then by Lemma 10, we have

$$\mathbb{E}_{s\sim\nu^*}\Big[V_\tau^{\pi_k}(s) - V_\tau^{\bar{\pi}_{k+1}}(s)\Big] = \mathbb{E}_{s\sim\nu^*}\Big[V_\tau^{\pi_k}(s) - V_\tau^{\pi_{k+1}}(s) + V_\tau^{\pi_{k+1}}(s) - V_\tau^{\bar{\pi}_{k+1}}(s)\Big]$$

$$\geq \mathbb{E}_{s\sim\nu^*}\Big[V_\tau^{\pi_k}(s) - V_\tau^{\pi_{k+1}}(s)\Big] - \big(\tfrac{B}{1-\gamma}\delta + \tfrac{C}{1-\gamma}\tau\big). \tag{44}$$

Note that we have assume the action space is bounded by $R$ and $\pi_{k+1}$ satisfy $T_2$

$$\big|\mathsf{W}_2^2(\bar{\pi}_{k+1}(\cdot|s), \pi^*(\cdot|s)) - \mathsf{W}_2^2(\pi_{k+1}(\cdot|s), \pi^*(\cdot|s))\big|$$

$$= \big|\big(\mathsf{W}_2(\bar{\pi}_{k+1}, \pi^*) - \mathsf{W}_2(\pi_{k+1}, \pi^*)\big)\big(\mathsf{W}_2(\bar{\pi}_{k+1}, \pi^*) + \mathsf{W}_2(\pi_{k+1}, \pi^*)\big)\big|$$

$$\leq \sqrt{2}R\,\mathsf{W}_2(\bar{\pi}_{k+1}(\cdot|s), \pi_{k+1}(\cdot|s)) \tag{45}$$

$$\leq 4R\sqrt{\tfrac{1}{\lambda}\delta}.$$

Hence,

$$\mathsf{W}_2^2(\pi_{k+1}(\cdot|s), \pi^*(\cdot|s)) \geq \mathsf{W}_2^2(\bar{\pi}_{k+1}(\cdot|s), \pi^*(\cdot|s)) - 4R\sqrt{\tfrac{1}{\lambda}\delta}. \tag{46}$$

Similarly,

$$\big|\mathsf{W}_2^2(\pi_{k+1}(\cdot|s), \pi_k(\cdot|s)) - \mathsf{W}_2^2(\bar{\pi}_{k+1}(\cdot|s), \pi_k(\cdot|s))\big| \leq 4R\sqrt{\tfrac{1}{\lambda}\delta}. \tag{47}$$

Combining (44), (46), and (47), we obtain

$$\mathbb{E}_{s\sim\nu^*}\Big[(1-\gamma)\big(V_\tau^{\pi^*}(s) - V_\tau^{\pi_k}(s)\big)\Big] + \mathbb{E}_{s\sim\nu^*}\Big[V_\tau^{\pi_k}(s) - V_\tau^{\pi_{k+1}}(s)\Big]$$

$$+ \mathbb{E}_{s\sim\nu^*}\Big[\tfrac{1}{2\eta_k}\mathsf{W}_2^2\big(\pi_k(\cdot|s), \pi_{k+1}(\cdot|s)\big)\Big]$$

$$\leq \mathbb{E}_{s\sim\nu^*}\Big[\tfrac{1}{2\eta_k}\mathsf{W}_2^2\big(\pi_k(\cdot|s), \pi^*(\cdot|s)\big) - \tfrac{\lambda\tau}{2}\mathsf{W}_2^2\big(\pi_{k+1}(\cdot|s), \pi^*(\cdot|s)\big)\Big]$$

$$+ \Big(\tfrac{B}{1-\gamma} + \tfrac{2R}{\eta_k}\sqrt{\tfrac{1}{\lambda}} + 2R\tau\sqrt{\lambda}\Big)\delta + \tfrac{C}{1-\gamma}\tau. \tag{48}$$

By rewriting

$$V_\tau^{\pi_k}(s) - V_\tau^{\pi_{k+1}}(s) = V_\tau^{\pi_k}(s) - V_\tau^{\pi^*}(s) + V_\tau^{\pi^*}(s) - V_\tau^{\pi_{k+1}}(s),$$

and rearranging the inequality, we have

$$\mathbb{E}_{s\sim\nu^*}\Big[V_\tau^{\pi^*}(s) - V_\tau^{\pi_{k+1}}(s)\Big] + \lambda\tau\,\mathbb{E}_{s\sim\nu^*}\Big[\tfrac{1}{2}\mathsf{W}_2^2\big(\pi_{k+1}(\cdot|s), \pi^*(\cdot|s)\big)\Big]$$

$$+ \mathbb{E}_{s\sim\nu^*}\Big[\tfrac{1}{2}\mathsf{W}_2^2(\pi_k(\cdot|s), \pi_{k+1}(\cdot|s))\Big]$$

$$\leq \gamma\,\mathbb{E}_{s\sim\nu^*}\Big[V_\tau^{\pi^*}(s) - V_\tau^{\pi_k}(s) + \tfrac{1}{2\eta_k\gamma}\mathsf{W}_2^2(\pi_k(\cdot|s), \pi^*(\cdot|s))\Big]$$

$$+ \Big(\tfrac{B}{1-\gamma} + \tfrac{2R}{\eta_k}\sqrt{\tfrac{1}{\lambda}} + 2R\tau\sqrt{\lambda}\Big)\delta + \tfrac{C}{1-\gamma}\tau.$$

Thus,

$$\mathbb{E}_{s\sim\nu^*}\Big[V_\tau^{\pi^*}(s) - V_\tau^{\pi_{k+1}}(s) + \tfrac{\lambda\tau}{2}\mathsf{W}_2^2(\pi_{k+1}(\cdot|s), \pi^*(\cdot|s))\Big]$$

$$\leq \gamma\,\mathbb{E}_{s\sim\nu^*}\Big[V_\tau^{\pi^*}(s) - V_\tau^{\pi_k}(s) + \tfrac{1}{2\eta_k\gamma}\mathsf{W}_2^2(\pi_k(\cdot|s), \pi^*(\cdot|s))\Big]$$

$$+ \Big(\tfrac{B}{1-\gamma} + \tfrac{R}{\eta_k}\sqrt{\tfrac{2}{\lambda}} + 2R\tau\sqrt{2\lambda}\Big)\delta + \tfrac{C}{1-\gamma}\tau.$$

Recalling the definitions of $J$ (9) and $\mathcal{D}$, we obtain

$$J(\pi^*)-J(\pi_{k+1})+\lambda\tau\mathcal{D}(\pi_{k+1},\pi^*) \leq \gamma\Big[J(\pi^*)-J(\pi_k)+\tfrac{1}{\eta_k\gamma}\mathcal{D}(\pi_k,\pi^*)\Big]+\Big(\tfrac{B}{1-\gamma}+\tfrac{2R}{\eta_k}\sqrt{\tfrac{1}{\lambda}}+2R\tau\sqrt{\lambda}\Big)\delta+\tfrac{C}{1-\gamma}\tau.$$
(49)

Choosing $\eta_k = \eta \geq \frac{1}{\gamma\lambda\tau}$ in the JKO scheme, we obtain

$$J(\pi^*) - J(\pi_{k+1}) + \lambda\tau\mathcal{D}(\pi_{k+1},\pi^*) \leq \gamma\Big[J(\pi^*) - J(\pi_k) + \lambda\tau\mathcal{D}(\pi_k,\pi^*)\Big]$$
$$+ \Big(\tfrac{B}{1-\gamma} + 3R\tau\sqrt{2\lambda}\Big)\delta + \tfrac{C}{1-\gamma}\tau,$$

which implies

$$J(\pi^*) - J(\pi_k) + \lambda\tau\mathcal{D}(\pi_k,\pi^*) \leq \gamma^k\Big[J(\pi^*) - J(\pi_0) + \lambda\tau\mathcal{D}(\pi_0,\pi^*)\Big] + \mathcal{O}(\delta + \tau).$$
$\square$

For the ease of presentation, we denote $\Delta_k = Q^{\pi_k,\xi_k} - Q^{\pi_k}$ and $\xi_{0:k} = \{\xi_0, \xi_1, \cdots, \xi_k\}$ in the following paper.

**Lemma 11.** *Under Assumption 2, for any state $s$ we have:*

$$\mathbb{E}_{\xi_{0:k}}[\langle\Delta_k(\cdot,s),\pi_{k+1}(\cdot|s)-\pi_k(\cdot|s)\rangle] \leq 2\eta_k\sigma_k^2 + \frac{1}{2\eta_k}\mathbb{E}_{\xi_{0:k}}\mathsf{W}_2^2(\pi_k(\cdot|s),\pi_{k+1}(\cdot|s))$$
(50)

*Proof.* For any $s$, let $\gamma(a,a'|s)$ be the optimal couple of the two distribution $\pi_k(\cdot|s)$ and $\pi_{k+1}(\cdot|s)$ in $\mathsf{W}_2$.

$$\mathbb{E}_{\xi_{0:k}}[\langle\Delta_k(a,s),\pi_{k+1}(a|s)-\pi_k(a|s)\rangle|\xi_{0:k-1}]$$
$$= \mathbb{E}_{\xi_k}[\int_{\mathcal{A}}\Delta_k(\cdot,s)\mathrm{d}(\pi_{k+1}(\cdot|s)-\pi_k(\cdot|s))|\xi_{0:k-1}]$$
$$= \mathbb{E}_{\xi_{0:k}}[\iint_{\mathcal{A}\times\mathcal{A}}\Delta_k(a,s)-\Delta_k(a',s)\mathrm{d}\gamma(a,a'|s)|\xi_{0:k-1}]$$
$$= \mathbb{E}_{\xi_{0:k}}[\iint_{\mathcal{A}\times\mathcal{A}}\int\langle\nabla_a\Delta_k((1-t)a'+ta,s),a-a'\rangle\mathrm{d}t\mathrm{d}\gamma(a,a'|s)|\xi_{0:k-1}]$$
$$= \iint_{\mathcal{A}\times\mathcal{A}}\int\mathbb{E}_{\xi_{0:k}}[\langle\nabla_a\Delta_k((1-t)a'+ta,s),a-a'\rangle|\xi_{0:k-1}]\mathrm{d}t\mathrm{d}\gamma(a,a'|s)$$
$$\leq \iint_{\mathcal{A}\times\mathcal{A}}\int\mathbb{E}_{\xi_{0:k}}[2\eta_k\|\nabla_a\Delta_k((1-t)a'+ta,s)\|_2^2+\frac{1}{2\eta_k}\|a-a'\|_2^2|\xi_{0:k-1}]\mathrm{d}t\mathrm{d}\gamma(a,a'|s)$$
$$\leq 2\eta_k\sigma_k^2 + \frac{1}{2\eta_k}\mathbb{E}_{\xi_{0:k}}[\mathsf{W}_2^2(\pi_k(\cdot|s),\pi_{k+1}(\cdot|s))|\xi_{0:k-1}]$$

The second equality applies the definition of an optimal coupling $\gamma(\cdot,\cdot|s) \in \Gamma(\pi_{k+1}(\cdot|s),\pi_k(\cdot|s))$, which means has the same marginal distribution as $\pi_k$ and $\pi_{k+1}$. The second inequality uses Young's inequality $\langle u,v\rangle \leq \frac{1}{2\eta_k}\|u\|^2 + \frac{\eta_k}{2}\|v\|^2$ to separate the two terms. The last inequality bounds the variance term of the stochastic gradient by $\sigma_k^2$ yields the last inequality, where the quadratic term recovers the squared Wasserstein distance between $\pi_k(\cdot|s)$ and $\pi_{k+1}(\cdot|s)$.

Taking expectation with respect to $\xi_{0:k-1}$ on both sides, we have the final result:

$$\mathbb{E}_{\xi_{0:k}}[\langle\Delta_k(\cdot,s),\pi_{k+1}(\cdot|s)-\pi_k(\cdot|s)\rangle] \leq 2\eta_k\sigma_k^2 + \frac{1}{2\eta_k}\mathbb{E}_{\xi_{0:k}}\mathsf{W}_2^2(\pi_k(\cdot|s),\pi_{k+1}(\cdot|s))$$
$\square$

**Theorem 2** (Linear Convergence). *Suppose Assumptions 1 and 2 hold, and for all $k \geq 0$, $\epsilon_k \leq \epsilon$, $\sigma_k \leq \sigma$ and $\|Q^{\pi_k,\xi_k}\|_\infty \leq B$. Then the iterates of (11) using step size $\eta_k = \eta \geq \frac{1}{\gamma\lambda\tau}$ satisfies*

$$\mathbb{E}_{\xi_{0:k-1}}\Big[J(\pi^*) - J(\pi_k) + \lambda\tau D(\pi_k,\pi^*)\Big] \leq \gamma^k\Big[J(\pi^*) - J(\pi_0) + \lambda\tau D(\pi_0,\pi^*)\Big] + \mathcal{O}(\delta + \tau + \epsilon + \sigma)$$
(15)

*where $J$ is defined in (9), and $\mathcal{D}(\pi_k,\pi^*) := \mathbb{E}_{s\sim\nu^*}\left[\frac{1}{2}\mathsf{W}_2^2\big(\pi_k(\cdot|s),\pi^*(\cdot|s)\big)\right]$. Consequently, in order to achieve an error of $\mathcal{O}(\varepsilon + \tau + \delta + \epsilon + \sigma)$ in expectation, the required iteration complexity*

*is*

$$\mathcal{O}\left(\frac{1}{1-\gamma}\,\log\frac{J(\pi^*)-J(\pi_0)+\lambda\tau\mathcal{D}(\pi_0,\pi^*)}{\varepsilon}\right).$$

*Proof.* By Lemma 8 applied to 11 with $p = \pi^*$, we have

$$\langle Q_\tau^{\pi_k,\xi_k}(s,\cdot),\pi^*(\cdot|s)-\bar\pi_{k+1}(\cdot|s)\rangle-\tau\mathsf{H}^{\pi^*}(s)+\tau\mathsf{H}^{\bar\pi_{k+1}}(s)+\frac{1}{2\eta_k}\mathsf{W}_2^2\big(\bar\pi_{k+1}(\cdot|s),\pi_k(\cdot|s)\big)$$

$$\le\frac{1}{2\eta_k}\mathsf{W}_2^2\big(\pi_k(\cdot|s),\pi^*(\cdot|s)\big)-\frac{\lambda\tau}{2}\mathsf{W}_2^2\big(\bar\pi_{k+1}(\cdot|s),\pi^*(\cdot|s)\big). \tag{51}$$

By Lemma 8 applied to 11 with $p = \pi_k$, we have

$$\Big(\langle Q_\tau^{\pi_k,\xi_k}(s,\cdot),\pi_k(\cdot|s)-\bar\pi_{k+1}(\cdot|s)\rangle-\tau\mathsf{H}^{\pi_k}(s)+\tau\mathsf{H}^{\bar\pi_{k+1}}(s)\Big)+\frac{1}{2\eta_k}\mathsf{W}_2^2\big(\bar\pi_{k+1}(\cdot|s),\pi_k(\cdot|s)\big)$$

$$\le-\frac{\lambda\tau}{2}\mathsf{W}_2^2\big(\bar\pi_{k+1}(\cdot|s),\pi^*(\cdot|s)\big)\le0.$$

Which implies that

$$\mathbb{E}_{s'\sim d_s^{\pi_k}}\Big[\langle Q_\tau^{\pi_k,\xi_k}(s',\cdot),\bar\pi_{k+1}(\cdot|s')-\pi_k(\cdot|s')\rangle-\tau\mathsf{H}^{\bar\pi_{k+1}}(s')+\tau\mathsf{H}^{\pi_k}(s')+\frac{1}{2\eta_k}\mathsf{W}_2^2\big(\bar\pi_{k+1}(\cdot|s),\pi_k(\cdot|s)\big)\Big]$$

$$\le d_s^{\pi_k}(s)\Big[\langle Q_\tau^{\pi_k}(s,\cdot),\bar\pi_{k+1}(\cdot|s)-\pi_k(\cdot|s)\rangle-\tau\mathsf{H}^{\bar\pi_{k+1}}(s)+\tau\mathsf{H}^{\pi_k}(s)+\frac{1}{2\eta_k}\mathsf{W}_2^2\big(\bar\pi_{k+1}(\cdot|s),\pi_k(\cdot|s)\big)\Big]$$

$$\le(1-\gamma)\Big[\langle Q_\tau^{\pi_k}(s,\cdot),\bar\pi_{k+1}(\cdot|s)-\pi_k(\cdot|s)\rangle-\tau\mathsf{H}^{\bar\pi_{k+1}}(s)+\tau\mathsf{H}^{\pi_k}(s)+\frac{1}{2\eta_k}\mathsf{W}_2^2\big(\bar\pi_{k+1}(\cdot|s),\pi_k(\cdot|s)\big)\Big] \tag{52}$$

where the last inequality follows from the fact that $d_s^{\pi_k}(s)\ge(1-\gamma)$ due to the definition of $d_s^{\pi_k}$ and $s_0 = s$ with probability one.

Note that we can still use the performance difference identity 38

$$V_\tau^{\bar\pi_{k+1}}(s)-V_\tau^{\pi_k}(s)=\frac{1}{1-\gamma}\mathbb{E}_{s'\sim d_s^{\pi_k}}\Big[\langle Q_\tau^{\pi_k}(s',\cdot),\bar\pi_{k+1}(\cdot|s')-\pi_k(\cdot|s')\rangle-\tau\mathsf{H}^{\bar\pi_{k+1}}(s')+\tau\mathsf{H}^{\pi_k}(s')\Big]$$

$$=\frac{1}{1-\gamma}\mathbb{E}_{s'\sim d_s^{\pi_k}}\Big[\langle Q_\tau^{\pi_k,\xi_k}(s',\cdot),\bar\pi_{k+1}(\cdot|s')-\pi_k(\cdot|s')\rangle-\tau\mathsf{H}^{\bar\pi_{k+1}}(s')+\tau\mathsf{H}^{\pi_k}(s')$$

$$-\langle\Delta_k(\cdot,s'),\bar\pi_{k+1}(\cdot|s')-\pi_k(\cdot|s')\rangle\Big] \tag{53}$$

By multiplying both sides by -1 and taking expectation with respect to $\xi_{0:k}$ gives

$$\mathbb{E}_{\xi_{0:k}}\big[V_\tau^{\pi_k}(s)-V_\tau^{\bar\pi_{k+1}}(s)\big]$$

$$\le\frac{1}{1-\gamma}\mathbb{E}_{\xi_{0:k}}\mathbb{E}_{s'\sim d_s^{\pi_k}}\Big[\langle Q_\tau^{\pi_k,\xi_k}(s',\cdot),\pi_k(\cdot|s')-\bar\pi_{k+1}(\cdot|s')\rangle-\tau\mathsf{H}^{\pi_k}(s')+\tau\mathsf{H}^{\bar\pi_{k+1}}(s')$$

$$+\tfrac{1}{2\eta_k}\mathsf{W}_2^2\big(\pi_k(\cdot|s'),\bar\pi_{k+1}(\cdot|s')\big)\Big]+2\eta_k\sigma_k^2$$

$$\le\mathbb{E}_{\xi_{0:k}}\Big[\langle Q_\tau^{\pi_k}(s,\cdot),\pi_k(\cdot|s)-\bar\pi_{k+1}(\cdot|s)\rangle-\tau\mathsf{H}^{\pi_k}(s)+\tau\mathsf{H}^{\bar\pi_{k+1}}(s)$$

$$+\tfrac{1}{2\eta_k}\mathsf{W}_2^2\big(\bar\pi_{k+1}(\cdot|s),\pi_k(\cdot|s)\big)\Big]+\tfrac{2\eta_k\sigma_k^2}{1-\gamma}. \tag{54}$$

Taking expectation with $\xi_{0:k}$ on 51 and combine with 54, we have:

$$\mathbb{E}_{\xi_{0:k}}\big[\langle Q_\tau^{\pi_k,\xi_k}(s,\cdot),\pi_k(\cdot|s)-\pi^\star(\cdot|s)\rangle+\tau\mathsf{H}^{\pi_k}(s)-\tau\mathsf{H}^{\pi^*}(s)+V_\tau^{\bar\pi_{k+1}}(s)-V_\tau^{\pi_k}(s)\big]$$

$$\le\mathbb{E}_{\xi_{0:k}}\Big[\frac{1}{2\eta_k}\mathsf{W}_2^2(\pi_k(\cdot|s),\pi^*(\cdot|s))-\frac{\lambda\tau}{2}\mathsf{W}_2^2(\bar\pi_{k+1}(\cdot|s),\pi^*(\cdot|s))\Big]+\frac{2\eta_k\sigma_k^2}{1-\gamma}.$$

Finally, averaging over the distribution $s\sim\nu^*$ and noting that $s$ and $\xi_{0:k}$ are independent, we have

$$\mathbb{E}_{s\sim\nu^*,\xi_{0:k}}\big[\langle Q_\tau^{\pi_k,\xi_k}(s,\cdot),\pi^*(\cdot|s)-\pi_k(\cdot|s)\rangle-\tau\mathsf{H}^{\pi^*}(s)+\tau\mathsf{H}^{\pi_k}(s)+V_\tau^{\pi_k}(s)-V_\tau^{\bar\pi_{k+1}}(s)\big]$$

$$\le\mathbb{E}_{s\sim\nu^*,\xi_{0:k}}\Big[\frac{1}{2\eta_k}\mathsf{W}_2^2(\pi_k(\cdot|s),\pi^*(\cdot|s))-\frac{\lambda\tau}{2}\mathsf{W}_2^2(\bar\pi_{k+1}(\cdot|s),\pi^*(\cdot|s))\Big]+\frac{2\eta_k\sigma_k^2}{1-\gamma}. \tag{55}$$

Noting that

$$
\mathbb{E}_{\xi_k}\left[\langle Q_\tau^{\pi_k,\xi_k}(s,\cdot), \pi^*(\cdot|s) - \pi_k(\cdot|s)\rangle \,\middle|\, \xi_{0:k-1}\right]
$$

$$
= \mathbb{E}_{\xi_k}\Big[\langle Q_\tau^{\pi_k}(s,\cdot), \pi^*(\cdot|s) - \pi_k(\cdot|s)\rangle + \langle \bar{Q}_\tau^{\pi_k}(s,\cdot) - Q_\tau^{\pi_k}(s,\cdot), \pi^*(\cdot|s) - \pi_k(\cdot|s)\rangle
$$

$$
\qquad + \langle Q_\tau^{\pi_k,\xi_k}(s,\cdot) - \bar{Q}_\tau^{\pi_k}(s,\cdot), \pi^*(\cdot|s) - \pi_k(\cdot|s)\rangle \,\middle|\, \xi_{0:k-1}\Big] \tag{56}
$$

$$
\geq \langle Q_\tau^{\pi_k}(s,\cdot), \pi^*(\cdot|s) - \pi_k(\cdot|s)\rangle - 2\epsilon_k
$$

The first equality expands $Q_\tau^{\pi_k,\xi_k}$ into its expectation $Q_\tau^{\pi_k}$ plus two error terms, namely the bias $\bar{Q}_\tau^{\pi_k} - Q_\tau^{\pi_k}$ and the stochastic fluctuation $Q_\tau^{\pi_k,\xi_k} - \bar{Q}_\tau^{\pi_k}$. Taking conditional expectation w.r.t. $\xi_k$ eliminates the mean of the fluctuation term. Finally, using the uniform error bound $\|\bar{Q}_\tau^{\pi_k} - Q_\tau^{\pi_k}\|_\infty \leq \epsilon_k$ and noting that both $\pi_k(\cdot|s)$ and $\pi^*(\cdot|s)$ are probability measures (which implies $\|\pi^*(\cdot|s) - \pi_k(\cdot|s)\|_1 \leq 2$), Hölder's inequality yields the desired bound.

Combining 55 and 56 and using Lemma 6,

$$
\mathbb{E}_{s\sim\nu^*,\xi_{0:k}}\Big[(1-\gamma)\big(V_\tau^{\pi_k}(s) - V_\tau^{\pi^*}(s)\big) + V_\tau^{\bar{\pi}_{k+1}}(s) - V_\tau^{\pi_k}(s)\Big]
$$

$$
\leq \mathbb{E}_{s\sim\nu^*,\xi_{0:k}}\Big[\frac{1}{2\eta_k}\mathsf{W}_2^2(\pi_k(\cdot|s), \pi^*(\cdot|s)) - \frac{\lambda\tau}{2}\mathsf{W}_2^2(\bar{\pi}_{k+1}(\cdot|s), \pi^*(\cdot|s))\Big] + 2\epsilon_k + \frac{2\eta_k\sigma_k^2}{1-\gamma}. \tag{57}
$$

Then by 10, we have

$$
|V_\tau^{\pi_k}(s) - V_\tau^{\pi^*}(s)| \leq \mathcal{O}(\delta + \tau) \tag{58}
$$

And Note that the action space is bounded by $R$ and $\pi_{k+1}$ satisfy $T_2$

$$
\big|\mathsf{W}_2^2(\bar{\pi}_{k+1}(\cdot|s), \pi^*(\cdot|s)) - \mathsf{W}_2^2(\pi_{k+1}(\cdot|s), \pi^*(\cdot|s))\big|
$$

$$
= \big|\big(\mathsf{W}_2(\bar{\pi}_{k+1}, \pi^*) - \mathsf{W}_2(\pi_{k+1}, \pi^*)\big)\big(\mathsf{W}_2(\bar{\pi}_{k+1}, \pi^*) + \mathsf{W}_2(\pi_{k+1}, \pi^*)\big)\big| \tag{59}
$$

$$
\leq \sqrt{2}R\,\mathsf{W}_2(\bar{\pi}_{k+1}(\cdot|s), \pi_{k+1}(\cdot|s))
$$

$$
\leq \mathcal{O}(\delta).
$$

combining (57), (58), (59), we have

$$
\mathbb{E}_{s\sim\nu^*,\xi_{0:k}}\Big[(1-\gamma)\big(V_\tau^{\pi_k}(s) - V_\tau^{\pi^*}(s)\big) + V_\tau^{\pi_{k+1}}(s) - V_\tau^{\pi_k}(s)\Big]
$$

$$
\leq \mathbb{E}_{s\sim\nu^*,\xi_{0:k}}\Big[\frac{1}{2\eta_k}\mathsf{W}_2^2(\pi_k(\cdot|s), \pi^*(\cdot|s)) - \frac{\lambda\tau}{2}\mathsf{W}_2^2(\pi_{k+1}(\cdot|s), \pi^*(\cdot|s))\Big]
$$

$$
+ \mathcal{O}(\delta + \tau) + 2\epsilon_k + \frac{2\eta_k\sigma_k^2}{1-\gamma}. \tag{60}
$$

Decomposing $V_\tau^{\pi_k}(s) - V_\tau^{\pi_{k+1}}(s)$ into $V_\tau^{\pi_k}(s) - V_\tau^{\pi^*}(s) - \big(V_\tau^{\pi_{k+1}}(s) - V_\tau^{\pi^*}(s)\big)$, recalling our definition of $J$ (9) and rearranging the terms in the above inequality, we get

$$
\mathbb{E}_{\xi_{0:k}}[J(\pi^*) - J(\pi_{k+1}) + \lambda\tau\mathcal{D}(\pi_{k+1}, \pi^\star)]
$$

$$
\leq \mathbb{E}_{\xi_{0:k-1}}[\gamma(J(\pi^*) - J(\pi_k)) + \frac{1}{\eta_k}\mathcal{D}(\pi^*, \pi_k)] + \mathcal{O}(\delta + \tau) + 2\epsilon_k + \frac{\eta_k\sigma_k^2}{2(1-\gamma)}. \tag{61}
$$

By choosing $\eta_k = \eta \geq \frac{1}{\gamma\lambda\tau}$, and for all $k \geq 0$, $\epsilon_k \leq \epsilon, \sigma_k \leq \sigma$, we get

$$
\mathbb{E}_{\xi_{0:k-1}}\Big[J(\pi^*) - J(\pi_k) + \lambda\tau D(\pi_k, \pi^*)\Big] \leq \gamma^k\Big[J(\pi^*) - J(\pi_0) + \lambda\tau D(\pi_0, \pi^*)\Big] + \mathcal{O}(\delta + \tau + \epsilon + \sigma) \tag{62}
$$

$$
\square
$$

# B NUMERICAL

## B.1 OVERALL ALGORITHM

**WPPG vs. WPPG-I: commonalities and differences** Both WPPG and WPPG-I are off-policy actor–critic methods built on the same backbone: (i) replay-based training with 1-step TD targets; (ii) Double-$Q$ critics with target networks and Polyak averaging; (iii) multi-sample bootstrap for target construction (average over $K$ next-action samples and take $\min(Q_1, Q_2)$); (iv) actor updates driven by action-gradient matching, i.e., aligning the policy's action increment with a noisy target direction, and (v) `tanh` squashing that maps actions to box constraints.

*Key difference.* WPPG employs an explicit Tanh–Gaussian policy $a = \mathrm{Affine}(\tanh(\mu_\theta(s) + \sigma_\theta(s) \odot \varepsilon))$ with a closed-form density (useful if one wishes to incorporate entropy/KL terms). WPPG-I uses a latent-conditioned *implicit* policy $a = \mathrm{Affine}(\tanh(f_\theta([s, z])))$, where $z \sim \mathcal{N}(0, I)$ is concatenated with the state; the policy distribution is implicit (no closed-form $\log \pi$), and learning relies purely on pathwise gradients through $\nabla_a Q$ and the shared latent variable reforwarding trick. Operationally, WPPG controls stochasticity via the Gaussian actor's output scale, whereas WPPG-I controls it via the *input* latent variables (its scale and dimensionality), enabling richer, state-conditional exploration.

---

**Algorithm 1** WPPG with Replay and Double-$Q$ Critics (Gaussian policy)

---

**Require:** Initialize actor $\pi_\theta(a|s) = \mathcal{N}(\mu_\theta(s), \Sigma_\theta(s))$ with Tanh squash; twin critics $Q_{w_1}, Q_{w_2}$ and targets $\bar{Q}_{w_1}, \bar{Q}_{w_2}$; target actor $\bar{\pi}_\theta$; replay buffer $\mathcal{D}$; step size $\eta$, noise scale $\tau$, samples per state $K$, discount $\gamma$, Polyak $\sigma$.

1: **for** each episode **do**
2:      Initialize $s_0$
3:      **for** $t = 0$ to $T - 1$ **do**
4:          Sample $a_t \sim \pi_\theta(\cdot|s_t)$, execute, observe $r_t, s_{t+1}$ and store $(s_t, a_t, r_t, s_{t+1})$ in $\mathcal{D}$
5:          **if** $\mathrm{len}(\mathcal{D}) \geq \mathrm{batch\_size}$ **then**
6:              Sample a minibatch $\{(s_i, a_i, r_i, s_i')\}_{i=1}^B$ from $\mathcal{D}$

7:          **Compute 1-step TD targets (multi-sample bootstrap using target nets):**
8:          For each $s_i'$, draw $\epsilon_{i,1:K} \sim \mathcal{N}(0, I)$ and set $a_{i,k}' \leftarrow \bar{\pi}_\theta(s_i'; \epsilon_{i,k})$
9:          $\hat{Q}_i \leftarrow \frac{1}{K} \sum_{k=1}^K \min(\bar{Q}_{w_1}(s_i', a_{i,k}'), \bar{Q}_{w_2}(s_i', a_{i,k}'))$
10:        $y_i \leftarrow r_i + \gamma \hat{Q}_i$

11:        **Critic update (train both critics):**
12:        $w_j \leftarrow w_j - \beta_Q \nabla_{w_j} \frac{1}{B} \sum_i (Q_{w_j}(s_i, a_i) - y_i)^2, \quad j \in \{1, 2\}$

13:        **Actor update (WPPG step with action-sample direction):**
14:        For each $s_i$, draw shared $\epsilon_{i,1:K}$ and form $a_{i,k} = \pi_\theta(s_i; \epsilon_{i,k})$; let $A_i = [a_{i,1:K}]$
15:        Compute $q_{i,k} = \min(Q_{w_1}(s_i, a_{i,k}), Q_{w_2}(s_i, a_{i,k}))$
16:        Obtain $\nabla_a Q$ at samples: $G_i = [\nabla_a q_{i,k}]_{k=1}^K$
17:        Form noisy target direction: $\Delta_i^\star \leftarrow \eta G_i + \xi_i$, where $\xi_i \sim \mathcal{N}(0, 2\tau\eta I)$
18:        Re-sample $A_i' = \pi_\theta(s_i; \epsilon_{i,1:K})$ with the *same* $\epsilon$ and define $\Delta_i \leftarrow A_i' - A_i$
19:        Update actor by matching directions: $\theta \leftarrow \theta - \beta_\pi \nabla_\theta \frac{1}{BK} \sum_{i,k} \|\Delta_{i,k} - \Delta_{i,k}^\star\|_2^2$

20:        **Target updates (Polyak):** $\bar{w}_j \leftarrow \sigma w_j + (1 - \sigma)\bar{w}_j, \ \bar{\theta} \leftarrow \sigma\theta + (1 - \sigma)\bar{\theta}, \ j \in \{1, 2\}$
21:        **end if**
22:        $s_{t+1} \leftarrow s'$
23:      **end for**
24: **end for**

---

---

**Algorithm 2** WPPG-I with Replay and Double-$Q$ Critics (Implicit Policy)

---

**Require:** Implicit actor $a = g_\theta(s, z)$ with Tanh squash ($z \sim \mathcal{N}(0, I_M)$); twin critics $Q_{w_1}, Q_{w_2}$ and targets $\bar{Q}_{w_1}, \bar{Q}_{w_2}$; target actor $\bar{g}_\theta$; replay buffer $\mathcal{D}$; step size $\eta$, noise scale $\tau$, samples per state $K$, discount $\gamma$, Polyak $\sigma$.

1: **for** each episode **do**
2:     Initialize $s_0$
3:     **for** $t = 0$ to $T - 1$ **do**
4:         Sample $z_t \sim \mathcal{N}(0, I_M)$, set $a_t = g_\theta(s_t, z_t)$, step env, observe $(r_t, s_{t+1}, d_t)$
5:         Store $(s_t, a_t, r_t, s_{t+1}, d_t)$ into $\mathcal{D}$
6:         **if** $\text{len}(\mathcal{D}) \geq \text{batch\_size}$ **then**
7:             Sample a minibatch $\{(s_i, a_i, r_i, s_i', d_i)\}_{i=1}^{B}$ from $\mathcal{D}$

8:             **Compute 1-step TD targets (multi-sample bootstrap):**
9:             For each $s_i'$, draw $z_{i,1:K}' \sim \mathcal{N}(0, I_M)$ and set $a_{i,k}' \leftarrow \bar{g}_\theta(s_i', z_{i,k}')$
10:           $\hat{Q}_i \leftarrow \frac{1}{K} \sum_{k=1}^{K} \min(\bar{Q}_{w_1}(s_i', a_{i,k}'), \bar{Q}_{w_2}(s_i', a_{i,k}'))$
11:           $y_i \leftarrow r_i + \gamma(1 - d_i)\hat{Q}_i$

12:             **Critic update (train both critics):**
13:           $w_j \leftarrow w_j - \beta_Q \nabla_{w_j} \frac{1}{B} \sum_i (Q_{w_j}(s_i, a_i) - y_i)^2, \quad j \in \{1, 2\}$

14:             **Actor update (direction matching with shared noise):**
15:           For each $s_i$, draw shared $z_{i,1:K} \sim \mathcal{N}(0, I_M)$ and set $a_{i,k}^{(0)} \leftarrow g_\theta(s_i, z_{i,k})$
16:           Compute $q_{i,k} = \min(Q_{w_1}(s_i, a_{i,k}^{(0)}), Q_{w_2}(s_i, a_{i,k}^{(0)}))$
17:           Obtain $G_i = [\nabla_a q_{i,k}]_{k=1}^{K}$ at $a^{(0)}$ (stop grad to critics)
18:           Form target direction: $\Delta_i^\star \leftarrow \eta\, G_i + \xi_i$, where $\xi_i \sim \mathcal{N}(0, 2\tau\eta\, I)$
19:           Reforward with the *same* $z$: $a_{i,k}^{(1)} \leftarrow g_\theta(s_i, z_{i,k})$, define $\Delta_{i,k} \leftarrow a_{i,k}^{(1)} - a_{i,k}^{(0)}$
20:           Update actor: $\theta \leftarrow \theta - \beta_\pi \nabla_\theta \frac{1}{BK} \sum_{i,k} \|\Delta_{i,k} - \Delta_{i,k}^\star\|_2^2$

21:             **Target updates (Polyak):** $\bar{w}_j \leftarrow \sigma w_j + (1 - \sigma)\bar{w}_j, \ \bar{\theta} \leftarrow \sigma\theta + (1 - \sigma)\bar{\theta}, \ j \in \{1, 2\}$
22:         **end if**
23:         $s_{t+1} \leftarrow s'$
24:     **end for**
25: **end for**

---

### B.2 IMPLEMENTATIONS

#### B.2.1 ACTOR AND POLICY

All our models and baselines are implemented under the standard actor–critic framework. Below we provide the implementation details for the actor and critic components separately.

**Action Squashing.** For consistency across methods, we apply a *tanh* squashing function to map sampled actions into the valid box $[a_{\min}, a_{\max}]$ for all algorithms. This squashing is crucial for the implicit policy: without it, when the injected-latent dimension is high, many actions are hard-clipped at the bounds, which prevents meaningful exploration and gradients, often leading to training failure. Below we present concise formulations of the two actors used.

**Tanh–Gaussian MLP Policy (used in WPPG/SAC/WPO).** Given state $s \in \mathbb{R}^S$, the actor outputs $\mu_\theta(s), \log \sigma_\theta(s) \in \mathbb{R}^A$ and samples
$$a = \frac{a_{\max} - a_{\min}}{2} \odot \tanh(\mu_\theta(s) + \sigma_\theta(s) \odot \varepsilon) + \frac{a_{\max} + a_{\min}}{2}, \qquad \varepsilon \sim \mathcal{N}(0, I_A),$$
i.e., a tanh-squashed Gaussian mapped to $[a_{\min}, a_{\max}]$ via an MLP producing $(\mu_\theta, \log \sigma_\theta)$.

**Noise-Conditioned Deterministic Policy (used in WPPG-I).** Given state $s \in \mathbb{R}^S$ and latent variables $z \in \mathbb{R}^M$,
$$a = \frac{a_{\max} - a_{\min}}{2} \odot \tanh(f_\theta([s, z])) + \frac{a_{\max} + a_{\min}}{2}, \qquad z \sim \mathcal{N}(0, I_M),$$
where $f_\theta$ is an MLP taking the concatenated input $[s, z]$. This defines an implicit policy (no closed-form density) with tanh-squashed outputs mapped to $[a_{\min}, a_{\max}]$.

### B.2.2 CRITIC

**Critic Learning Target** For all off-policy algorithms (SAC, WPPG, WPPG-I, WPO), the critic is trained with 1-step TD targets that average over $K$ bootstrap action samples and use Double-$Q$ when available:

$$y_t \;=\; r_t + \gamma(1-d_t)\,\frac{1}{K}\sum_{k=1}^{K}\min_{j\in\{1,2\}} Q_{\bar{w}_j}\big(s_{t+1}, a'_{t+1,k}\big), \qquad a'_{t+1,k} \;=\; g_{\bar{\theta}}\big(s_{t+1},\varepsilon_k\big), \;\; \varepsilon_k \sim \mathcal{N}(0, I).$$

Here, $j \in \{1,2\}$ indexes the two target critics used by Double-$Q$ (the per-sample minimum is taken), and the outer average is over $K$ target actions drawn from the target actor $g_{\bar{\theta}}$. For single-$Q$ methods (e.g., WPO *uses a single $Q$*, or the single-$Q$ WPPG ablation), replace $\min_{j\in\{1,2\}} Q_{\bar{w}_j}$ by $Q_{\bar{w}}$. In contrast, PPO retains its on-policy generalized advantage estimation (GAE) for actor updates.

## B.3 HYPERPARAMETERS

**Neural Network Architecture** All experiments are based on two standard network configurations: a larger network with hidden sizes (256, 256) and ReLU activation, and a smaller network with hidden sizes (64, 64) and Tanh activation. We use the larger network for Hopper, Humanoid, and HalfCheetah, and the smaller network for all other tasks. The same choice is applied uniformly across all models, and the actor and critic share the same network architecture.

**Replay Buffer** For consistency, all off-policy algorithms (SAC, WPPG, WPPG-I, WPO) use the same replay buffer configuration as summarized in Table 1, ensuring identical storage capacity, sampling scheme, and update frequency across methods.

**Training Setup** All off-policy models share the same basic training setup: each is trained for $1 \times 10^6$ timesteps, with evaluation performed every 2000 steps. Target networks are updated via Polyak averaging to stabilize critic training. Both actor and critic use a learning rate of $3 \times 10^{-4}$. The discount factor is set to $\gamma = 0.99$ for all tasks, except Swimmer where $\gamma = 0.9999$. The configuration is summarized in Table 2. For PPO, hyperparameters are environment-specific and detailed below.



Table 1: Replay buffer.

| Hyperparameter | Value |
|---|---|
| Buffer size | 1,000,000 |
| Batch size | 256 |
| Learning starts | 10,000 |
| Train frequency | 1 (step) |
| Gradient steps per update | 1 |
| Number of environments | 1 |

Table 2: Training Setup.

| Hyperparameter | Value |
|---|---|
| Discount factor $\gamma$ | 0.99 |
| Polyak coefficient | 0.005 |
| Learning rate (actor/critic) | $3 \times 10^{-4}$ |
| Target update interval | 1 |
| Total timesteps | 1,000,000 |
| Optimizer | Adam |



**Model Specific Hyperparameters** The hyperparameters of all models are summarized in the tables below. For WPO and SAC, we follow the settings reported in the WPO paper, while the hyperparameters of PPO are taken from RL Zoo. Although we find that tuning hyperparameters for each environment can often improve performance, for simplicity and fairness of comparison we adopt a single unified set of hyperparameters for all off-policy methods across all tasks, which yields stable and competitive learning performance.

## B.4 ADDITIONAL EXPERIMENT RESULTS

**Multi-Run Evaluation** To more comprehensively demonstrate the behavior of WPPG and WPPG-I, we further evaluate both methods with multiple training runs. Specifically, each algorithm is trained 5 times with different random seeds. In the corresponding plots, the solid line denotes the mean return across the 5 runs, while the shaded area indicates the range between the minimum and maximum returns over these runs (see Fig.3).

Table 3: WPPG-I hyperparameters.

| Hyperparameter | Default Value |
|---|---|
| Action samples | 32 |
| Step size $\eta$ | 10 |
| Noise scale $\tau$ | 0.00001 |
| Actor Latent Dimension | $\frac{1}{3} \times$ State Dimension |

Table 4: WPPG hyperparameters.

| Hyperparameter | Default Value |
|---|---|
| Action samples | 32 |
| Step size $\eta$ | 0.01 |
| Noise scale $\tau$ | 0.00001 |

Table 5: SAC hyperparameters.

| Hyperparameter | Default Value |
|---|---|
| Entropy Coefficient $\alpha$ | 0.001 |
| Maximum Policy Variance | $\exp(4)$ |
| Minimum Policy Variance | $\exp(-10)$ |

Table 6: WPO hyperparameters.

| Hyperparameter | Default Value |
|---|---|
| KL Mean Penalty $\alpha_\mu$ | 0.001 |
| KL Variance Penalty $\alpha_\Sigma$ | 0.001 |
| Action samples | 32 |

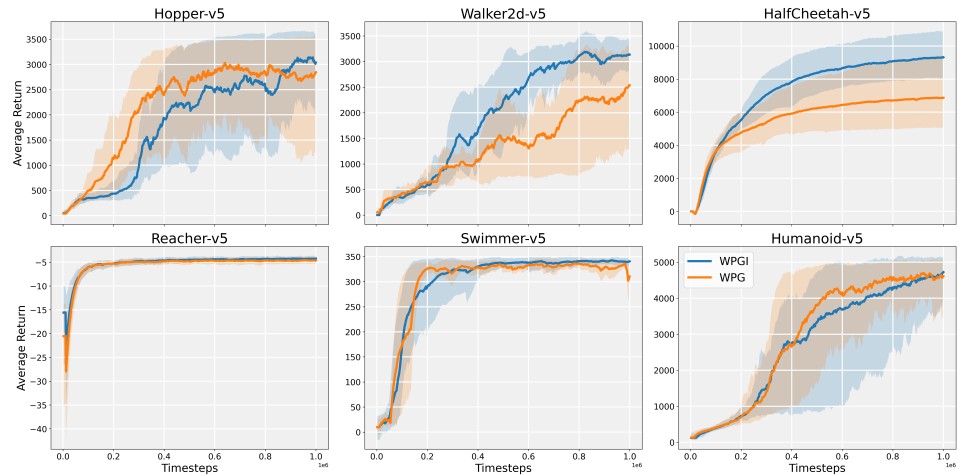

Figure 3: Multi-Run Evaluation

**Combined Humanoid Task**    Our method shares some similarities with SAC in that both are based on entropy regularization and use the action gradient of the Q-function for policy updates. However, the key advantage of WPPG lies in its ability to train an implicit policy. To better showcase this benefit, we follow the construction in WPO Pfau et al. (2025) and create a combined task that increases the action dimensionality: multiple Humanoid environments are run in parallel, their states are concatenated and fed into a single agent, which outputs the concatenated actions jointly. As shown in the combined Humanoid task, WPPG-I converges to consistently higher returns than SAC, indicating that the implicit policy is able to discover action distributions that achieve higher rewards. (see Fig.4).

**Ablation on Double Q Function**    Double-$Q$ plays a crucial role for WPPG. As shown in the figures, although the single-$Q$ variant of WPPG outperforms WPO on most environments, it fails to achieve fast and stable learning on challenging tasks such as Humanoid. Beyond stability, the use of double-$Q$ also opens up interesting directions for further exploration; for example, one could choose the $Q$-function with the smaller gradient magnitude to provide the action-sample update direction. We leave such extensions for future work.

**Additional Ablation Study**    Beyond the main results, we also conduct additional ablation studies on Humanoid-v5 and HalfCheetah-v5, systematically varying key hyperparameters (e.g., the Wasserstein step size $\eta$, the number of sampled actions, and the latent dimension of the implicit policy). These experiments, reported in the supplementary material, further validate the robustness of our method and illustrate how performance and stability depend on these design choices. (see Fig.6 and Fig.7).

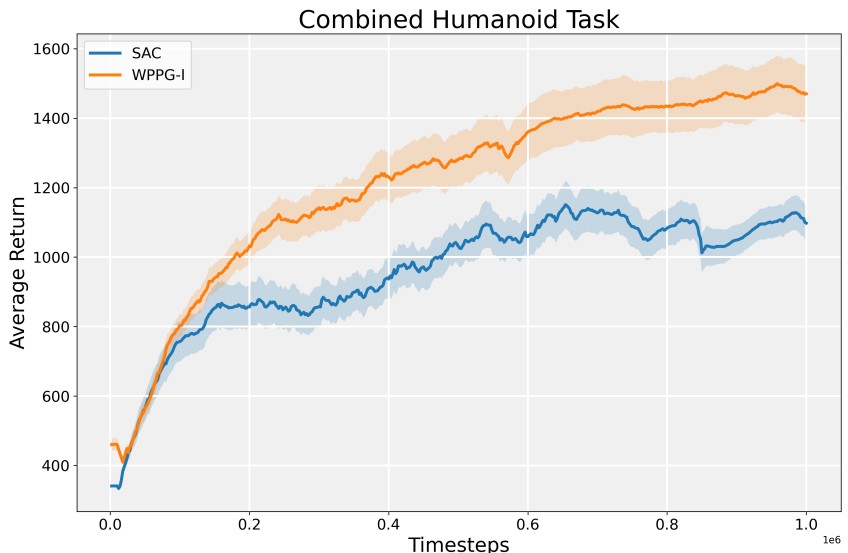

Figure 4: Combined Humanoid Task

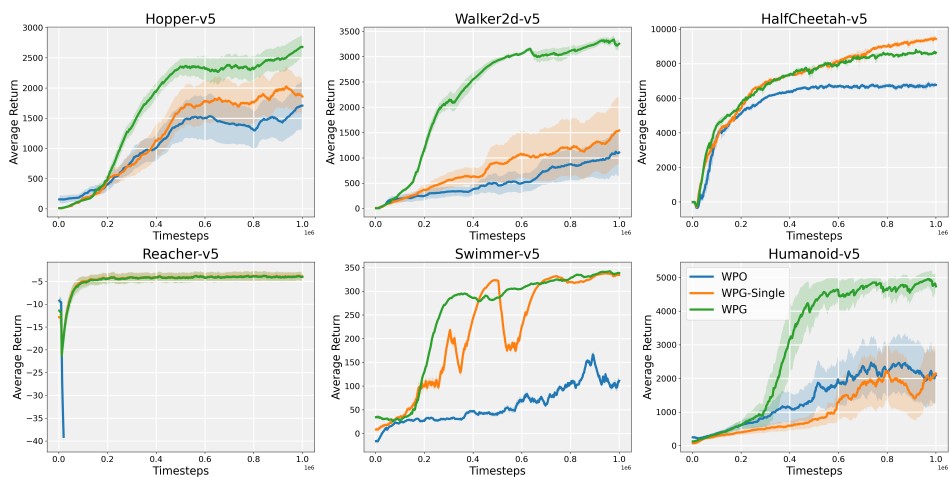

Figure 5: Ablation on Double Q Function

**Training Speed Comparison** All experiments are conducted on an NVIDIA RTX 4090 GPU. The training times on the high-dimensional Humanoid-v5 task are reported in Table 7. As can be seen, WPPG, WPPG-I, and WPO achieve comparable training speeds, all of which are faster than SAC.

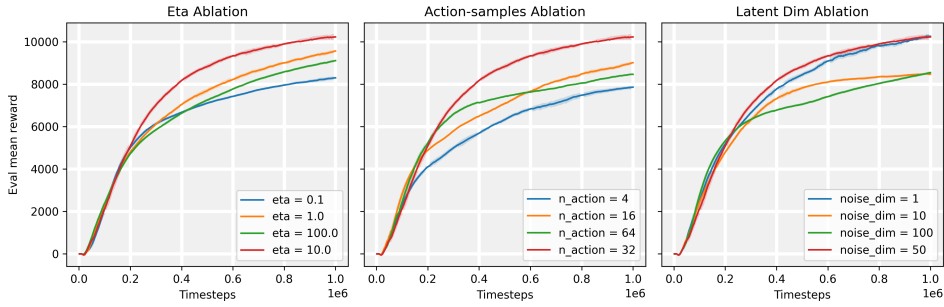

Figure 6: Additional Ablation on Eta, Action Samples and Latent Dim with HalfCheetah-v5 Task

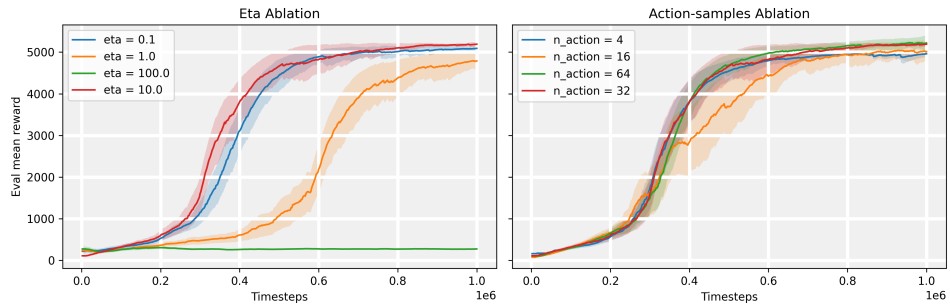

Figure 7: Additional Ablation on Eta, Action Samples and Latent Dim with Humanoid-v5 Task

Table 7: Training time on Humanoid-v5 (in seconds).

| Model | Time (s) |
|---|---|
| SAC | 19451 |
| WPPG-I | 12113 |
| WPPG | 13135 |
| WPO | 10649 |
| PPO | 2104 |

## C  ADDITIONAL DISCUSSION

We now discuss a practically relevant case where the action space $\mathcal{A}$ is convex and bounded. To ease the presentation, we fix a state $s$ in the discussion below, and all constants should be interpreted as a uniform constant for all states.

Consider the policy update

$$\rho_{k+1} \in \underset{\rho \in \mathcal{P}(A)}{\operatorname{argmin}} \left\{ \int_A Q_k(s,a)\rho(a)\, da + \tau \int_A \rho \log \rho\, da + \frac{1}{2\eta} W_2^2(\rho, \rho_k) \right\}.$$

Assume $Q_k$ has a uniformly bounded gradient. Assume the initial policy distribution $\rho_0$ satisfying
$$0 < m_0 \leq \rho_0(a) \leq M_0 < \infty \quad \text{for all } a \in \mathcal{A}. \tag{63}$$
Fix a continuous distribution $\nu$ on $\mathcal{A}$ with density $f$ such that
$$0 < \tilde{m} \leq f(a) \leq \tilde{M} < \infty \quad \text{for all } a \in \mathcal{A}. \tag{64}$$
The optimality condition yields
$$\frac{\rho_{k+1} - \rho_k}{\eta} = \tau \Delta \rho_{k+1} + \nabla \cdot (\rho_{k+1} \nabla Q) \quad \text{in } A, \tag{65}$$
with Neumann boundary condition
$$(\tau \nabla \rho_{k+1} + \rho_{k+1} \nabla Q) \cdot n = 0 \quad \text{on } \partial A. \tag{66}$$
By the maximum principle, if $\rho_k$ has uniformly positive lower and upper bounds, so does $\rho_{k+1}$. For each $k \geq 0$, let $T_k : \mathcal{A} \to \mathcal{A}$ be the Brenier map pushing $\nu$ to $\rho_k$. Then there exists a convex potential $\phi_k : \mathcal{A} \to \mathbb{R}$ such that
$$T_k(a) = \nabla \phi_k(a).$$

The convex gradient map $T_k = \nabla \phi_k$ solves the Monge-Ampère equation
$$\det D^2 \phi_k(x) = \frac{f(x)}{g_k(\nabla \phi_k(x))} \quad \text{for } x \in A.$$
Then by Caffarelli's regularity theory, $\phi_k$ has uniformly bounded Hessian. Therefore
$$\operatorname{Lip}(T_k) = \sup_{x \in A} \|DT_k(x)\| = \sup_{x \in A} \|D^2 \phi_k(x)\| \leq C \quad \text{for all } k \geq 0.$$

Thus, the family $\{T_k\}_{k \geq 0}$ of optimal transport maps is uniformly Lipschitz. As a direct consequence, if $\nu$ satisfies a $T_2$ inequality, then so do the JKO iterates $\rho_k$ with a uniform constant.

