# OpenReview forum: "Wasserstein Policy Gradient: Implicit Policies, Entropy Regularization and Linear Convergence"
_ICLR.cc/2026/Conference — Submitted to ICLR 2026_

### Official Review · Reviewer_zK6R · 2025-10-30

**Soundness:** 3
**Presentation:** 4
**Contribution:** 2
**Rating:** 4
**Confidence:** 3

**Summary:**

The paper proposes a new formulation of Wasserstein Policy Proximal Gradients with entropy regularization. The update rule can be applied to optimizing both explicit policies and implicit polices. The paper presents the convergence analysis when the update rule is applied to per-state maximization under exact and inexact Q estimate.  The paper also evaluates different variants of the proposed method (approximate version) on continuous control tasks.

**Strengths:**

### Originality
It’s novel to see that the proposed WPPG formulation can be applied to two different types of policies: the explicit parameterization with density functions and the implicit parameterization as state-to-action mapping. The empirical evaluation shows the method can training performant policies for both cases.

### Quality and clarity
The paper is well-written and easy to understand. All notations and equations are clearly defined and explained. The proposed methods have been rigorously studies from both the theoretical perspective and the empirical evaluation. The paper provides sufficient review on relevant literatures and clearly points out the difference and the gap.

**Weaknesses:**

### Novelty and Significance
The theoretical results are similar to those in existing publications, including Lan (2021) and Song et al. (2023). Also, these results are built on the tabular-like policy update rule, see Eq. (8) and (11), both of which differ greatly from the optimization of parameterized policies (as specified in Eq. (7)). Specifically, Eq. (8) and (11) are per-state policy update without any policy parameterization. Consequently, the linear convergence rate is applicable only to the exact per-state policy update, not even to the policies with implicit or explicit parameterization.

The empirical results do not show that the proposed WPPG and its variants are more performant than existing methods on the selected benchmark tasks. In fact, the WPPG and WPPG-I have the similar performance as the SAC on most learning environments. This is sensible since the WPPG and its variants leverage the same entropy regularization as used in SAC. This performance similarity is on the condition that the entropy regularization coefficient was fixed for SAC (“SAC is evaluated with entropy coefficient self-tuning disabled”), which might imply that the SAC could be more performant. The paper should at least present one learning environment that can clearly show the benefits of using WPPG.

**Questions:**

1. what’s a log-Sobolev condition on the policy class? It was mentioned in both Abstract and Conclusion but not explained in the main texts.

2. Line 236 – 238 is repetitive, same as the lines 233 – 235

3. Line 267, typo “assuption”

4. For Eq. (7), it only specifies the optimization for the implicit policies. What about the explicit policies?

5. As WPPG is derived based on the Q function, why the advantage function is then used in the formulation? Line 190.

---

> ### Author Response · Authors · 2025-11-26
>
> Thank you for the thoughtful feedback! Our detailed responses are as follows.
>
> **W1:** Theoretical results.
>
> Thank you for the thoughtful comparison and for raising this concern. We would like to clarify our key innovations and contributions more clearly, and why our theoretical results are not merely restatements of prior work nor limited to tabular updates.
>
> First, Lan’s linear-rate proof for mirror-decent policy update relies on the three-point identity of KL divergence, which does *not* hold in Wasserstein geometry. Hence, their argument cannot be transferred directly, even though our proof follows a similar high-level structure. Instead, our analysis develops new descent/contraction inequalities utilizing the properties of the Wasserstein metric and $T_2$ inequality. This is a key distinction, yielding a different proof and constants. To our knowledge, this gives the *first global linear (geometric) convergence* result for Wasserstein-based policy optimization.
>
> Second, the convergence result of Song et al.~applies only to finite action spaces, and their proof does not exploit any geometric properties of the Wasserstein distance, except for an upper bound on the diameter of the action space. In contrast, our Lemma 7 and Theorems 1–2 leverage the Wasserstein structure to obtain a nontrivial contraction, enabling a linear convergence rate.
>
> Third, although Eqs. (8) and (11) describe per-state proximal “oracle” updates, our guarantees extend beyond the tabular setting. We explicitly analyze the mismatch between the parametric policy and the oracle policy--captured by the realizability error $\delta$--and show that it contributes only an $O(\delta)$ error to the global convergence. Theorem 2 further accounts for critic approximation, establishing convergence in the inexact Q-function setting as well.
>
> **W2:** Benefits of using WPPG over SAC.
>
> Thank you for this insightful comment! To better highlight where WPPG brings a clear advantage, we added a combined multi-Humanoid task following the construction in the WPO paper: three Humanoid environments are run in parallel, their states are concatenated and fed into a shared agent, which outputs the concatenated action vector jointly. On this high-dimensional control problem, the implicit-policy variant WPPG-I converges to substantially higher returns than SAC, indicating that the implicit policy learned via WPPG can discover better action distributions in more complex settings. We refer the reviewer to revised Appendix B.4 (Combined Humanoid task) for these results.
>
> The similar performance in simpler tasks is partly our intended message. We aim to bridge Wasserstein policy optimization with established Q-value–based methods like SAC, showing that our implicit policy update can be implemented with competitive performance. Our most significant contribution, however, is the principled theoretical framework and convergence analysis that makes this implicit policy update possible.
>
> **Q1:** Log-Sobolev condition.
>
> Thank you for the question, and we apologize for the confusion. Our convergence proofs only assume the $T_2$ inequality. The log-Sobolev inequality was mentioned in the Abstract/Conclusion merely as a sufficient (stronger) condition that implies $T_2$. We have revised the paper to state the assumption directly in terms of $T_2$.
>
> **Q2 & Q3**.
>
> We have corrected the typos. Thanks!
>
> **Q4:** Explicit policies.
>
> In continuous action spaces, common explicit policies like Gaussian and Gaussian mixture models can be handled by our framework.
> On the other hand, since we specifically target implicit policies as a design choice, WPPG does not apply to arbitrary general explicit policies. Nonetheless, our new experiments validate the efficacy of implicit policies, and we anticipate their broader use in future work.
>
> **Q5:** Advantage function.
>
> We employ a standard variance reduction technique. Note that the update objective can be rewritten using the relation: $$\langle Q^\pi(s,\cdot), \pi(\cdot|s) \rangle = \langle A^\pi(s,\cdot), \pi(\cdot|s) \rangle + V^\pi(s)$$. Because $V^\pi(s)$ is independent of the policy variations at the current step, optimizing $Q$ and $A$ results in equivalent updates. We choose the advantage formulation because it maintains the same theoretical optimizer while significantly reducing variance.

---

### Official Review · Reviewer_qGJK · 2025-10-31

**Soundness:** 2
**Presentation:** 3
**Contribution:** 3
**Rating:** 4
**Confidence:** 2

**Summary:**

This paper introduces Wasserstein Proximal Policy Gradient (WPPG), a novel framework for policy optimization in reinforcement learning that performs proximal updates directly under Wasserstein geometry rather than the usual KL-based geometry. The proposed WPPG eliminates dependence on policy densities, enabling the optimization of implicit stochastic policies via gradient updates over action samples. Moreover, the authors derived linear convergence guarantees for both exact and approximate value functions. Empirically, the proposed WPPG outperforms standard baselines such as PPO, SAC, and WPO on MuJoCo benchmarks.

**Strengths:**

1. The paper provides a clear convergence proof of the proposed WPPG, adapting transport-information inequalities and entropy regularization.
2. The proposed WPPG enables learning in settings where policy density is intractable. And the projection-based update is simple yet powerful, avoiding the explicit computation of the log-density.
3. Empirically, WPPG-I consistently outperforms all baselines (PPO, SAC, WPO) on MuJoCo tasks.

**Weaknesses:**

1. The experiments are restricted to standard MuJoCo tasks. It would strengthen the paper to include additional environments (e.g., stochastic or discontinuous environments, sparse rewards) to demonstrate the robustness and generality of WPPG.
2. The computational overhead of introducing Wasserstein gradient flow is unclear. The paper briefly reports wall-clock time (Table 7) but does not provide detailed complexity comparisons (e.g., gradient computation cost per update vs. KL-based methods).
3. The effects of $\tau$ and latent dimension are demonstrated only on Humanoid. Moreover, additional ablation experiments on step size $\eta$, sample number K, critic depth, or replay buffer size would provide a clearer view of WPPG’s stability and sensitivity.
4. There are some typos that require further calibration, e.g., a duplicate paragraph appears at the beginning of Section 4.

**Questions:**

1. Can you provide a deeper theoretical or empirical comparison between the proposed WPPF and WPO? Since both methods use Wasserstein geometry but differ in projection (Wasserstein vs. KL). For example, whether WPPG achieves tighter bounds or better conditioning than WPO in continuous control?
2. Since WPPG optimizes actions using Wasserstein projection, the per-update cost may differ from KL-based methods. Can you provide empirical estimates of the complexity of WPPG updates relative to SAC/PPO/WPO?
3. In your convergence analysis, the log-Sobolev and $T_{2}$ assumptions are strong. Could the analysis be extended to weaker conditions?
4. Have you tested WPPG or WPPG-I on tasks beyond MuJoCo, such as stochastic environments (e.g., AntMaze, POMDPs, or hybrid control), to assess its broader applicability?
5. Can the approximate realizability parameter $\delta$ be quantified empirically, or does it vanish asymptotically as network capacity increases?

---

> ### Author Response · Authors · 2025-11-26
>
> Thank you for your valuable feedback! In the revision, we have addressed LaTeX issues. Please find our detailed responses below.
>
> **W1 & Q4:** Restricted experiments.
>
> We agree that focusing on MuJoCo is a limitation. Nonetheless, we would like to emphasize that the main contributions of this work are theoretical; the experiments serve as a proof-of-concept to demonstrate the algorithm's effectiveness. Our work provides the first global convergence result for Wasserstein-type policy gradients in continuous control. Our new Wasserstein policy gradient variant relies solely on the action gradient, analogous to standard Q-value–based policy optimization methods. As such, we choose MuJoCo continuous-control suite, the standard testbed for related benchmarks (including WPO [1]).
>
> To further strengthen our empirical results, we added a new experiment highlighting the advantages of our framework's support for implicit policies. Following the setup in WPO [1], we constructed a multi-Humanoid environment by combining several tasks in parallel: the agent receives concatenated states and outputs concatenated actions jointly. In this setting, our WPPG-I algorithm significantly outperforms SAC, demonstrating the benefits of an expressive implicit policy for complex control. While this submission focuses on MuJoCo, extending the evaluation to other benchmarks is an important direction for future work.
>
> **W2, Q1 and Q2:** Comparison with WPO/PPO/SAC.
>
> A primary distinction is that these methods (WPO, PPO, SAC) require evaluating or differentiating the policy's log density, which restricts common policy classes to Gaussians or Gaussian mixtures. In contrast, WPPG applies directly to implicit policies, such as an MLP pushforward. This extends the applicability of Wasserstein-based policy optimization beyond explicit distributions, and our newly added experiments demonstrate the efficacy of these implicit policies.
>
> Like SAC and WPO, our method WPPG utilizes the action gradient of the critic $Q$ to guide the update direction. Moreover, both WPPG (Eq. (7)) and WPO (Eq. (6) in [1]) involve a Newton-like update that can be solved using first-order methods or approximate second-order methods, such as KFAC or diagonal approximation. However, unlike WPO (Eq. (6) in [1]), WPPG does not compute the gradient of the policy's log density, thereby reducing computational overhead.
> Theoretically, assuming an accurate projection (so that the parameteric update acts as a faithful approximation of the distributional update in Wasserstein space), our theoretical result also provides a global convergence guarantee for WPO.
>
> **W3:** Ablation study.
>
> Thank you for pointing out the limitations in our ablation study. We agree that a more systematic analysis helps clarify the stability and sensitivity of WPPG. We refer the reviewer to the revised Appendix B.4 (Additional Ablations), where we have expanded our study to include step size $\eta$, sample number K, and latent dimension, evaluated on both Humanoid and HalfCheetah.
>
> Regarding critic depth and replay buffer size, we note that these hyperparameters are typically treated as *standard, fixed infrastructure parameters* in continuous-control RL benchmarks. Prior works such as SAC, TD3, and WPO all adopt stable, well-established configurations for these components and rarely consider them as algorithm-specific ablation variables. Varying them tends to test general neural-network capacity or replay-based RL stability rather than properties of the proposed algorithm itself. For this reason--and to maintain comparability with existing baselines--we follow the common practice of keeping critic depth and replay buffer size fixed across all methods. Nonetheless, we appreciate the reviewer’s perspective and consider broader architectural/hyperparameter explorations an interesting direction for future work.
>
> **Q3:** Log-Sobolev and $T_2$ assumptions.
>
> Thank you for this important question. We would like to clarify that our analysis only requires the $T_2$ inequality; the original version mentioned the log-Sobolev inequality (LSI) mainly as a sufficient route to $T_2$. We apologize for the confusion and have corrected the presentation accordingly.
>
> Regarding weaker assumptions, we are not aware of a substantially weaker condition--in fact, even in single-stage single-state problem, the existing results require LSI (e.g., [4]), which is stronger than $T_2$. Note that $T_2$ holds for practically relevant policy classes; see our response to Q5 for more details.
>
> For a bounded convex action space, any continuous policies with uniformly positive lower bound and upper bound satisfy a Talagrand $T_2$ inequality. Moreover, we show (see Appendix C) that for implicit policies parametrized by Lipschitz networks, if the initial policy satisfies this condition, then all policy iterates of our algorithm satisfy a uniform $T_2$ inequality. This validates the $T_2$ assumption in our setting.

---

> ### Author Response · Authors · 2025-11-26
> **Official Comment by Authors (Continued)**
>
> **Q5:** Approximate realizability.
>
> The approximate realizability parameter $\delta$ captures how well our chosen policy class can approximate the ideal Wasserstein proximal/oracle update. Importantly, $\delta$ is inherently task-dependent in RL: for simpler continuous-control tasks, a unimodal Gaussian policy may already approximate the oracle well, leading to a small $\delta$; whereas for more complex tasks with multimodal or hybrid action structure, richer classes (e.g., mixtures or implicit generators) may be required to make $\delta$ small. Although $\delta$ cannot be measured directly, it is a standard theoretical device for quantifying approximation error.
> Asymptotically, under standard universal approximation results for neural networks, $\delta$ is an approximation-error term that should decrease as network capacity increases. For example, for implicit policies parameterized by Lipschitz networks satisfy this assumption thanks to their universality [5][6].
>
>
>
> ### **References**
>
> [1] Pfau, David, et al. "Wasserstein Policy Optimization." International Conference on Machine Learning (2025).
>
> [2] Terpin, Antonio, et al. "Trust region policy optimization with optimal transport discrepancies: Duality and algorithm for continuous actions." Advances in Neural Information Processing Systems 35 (2022): 19786-19797.
>
> [3] Song, Jun, et al. "Provably Convergent Policy Optimization via Metric-aware Trust Region Methods." Transactions on Machine Learning Research 2023.6 (2023).
>
> [4] Chizat, Lénaïc. "Mean-Field Langevin Dynamics: Exponential Convergence and Annealing." Transactions on Machine Learning Research (2022).
>
> [5] Cem Anil, James Lucas, Roger Grosse. Sorting Out Lipschitz Function Approximation. Proceedings of the 36th International Conference on Machine Learning (ICML).
>
> [6] Davide Murari, Takashi Furuya, Carola-Bibiane Schönlieb. Approximation theory for 1-Lipschitz ResNets. arXiv preprint arXiv:2505.12003.

---

### Official Review · Reviewer_oEc8 · 2025-11-01

**Soundness:** 4
**Presentation:** 3
**Contribution:** 3
**Rating:** 8
**Confidence:** 3

**Summary:**

This paper considers policy optimization in distribution space, with specific attention to KL vs Wasserstein distance metrics. They propose Wasserstein Proximal Policy Gradient (WPPG), a policy optimization approach which eliminates needing access to policy densities or score functions and is readily applicable to optimizing implicit stochastic policies. They provide a linear convergence proof under entropy regularization and a log-Sobolev condition. They then evaluate WPPG and its implicit variant (WPPG-I) empirically.

**Strengths:**

* The paper provides good theoretical contributions in both WPPG's derivation and analysis, which—as far as I was able to verify—appear correct. For each main assumption at the start of 4.1, they provided details on how they might be met.

* The resulting algorithm is relatively easy to implement, which is not always the case when dealing with the Wasserstein metric. The sole dependence on the action-value gradient broadens its applicability to implicit policies and crucially opens a lot of avenues for further exploration.

* The paper is relatively well written and easy to follow, situates itself well among prior work, ad provides good motivation for considering the Wasserstein metric (e.g., respecting the geometry of the action space, intuitions behind earth-moving distance, etc.).

* The empirical evaluation provided relevant ablations to better understand the algorithm and its sensitivities.

**Weaknesses:**

* The empirical evaluation was relatively limited in that it only considered the MuJoCo suite. This isn't a huge concern though if the paper's primary contributions are theoretical.

* In the derivation of the Wasserstein projection, the shared-latent coupling is first-order optimal under small $\eta$. However, the convergence theorem lower bounds $\eta \geq \frac{1}{\gamma \lambda \tau}$, making the derivation and analysis appear at odds.

Minor which did not impact my review:

* The intro paragraph to Section 4 was duplicated.
* "under the geometry of optimal transport. (Pfau et al., 2025)." -> "under the geometry of optimal transport (Pfau et al., 2025)."

**Questions:**

* Can the authors comment on the seeming conflict between the derivation ideally having $\eta$ be small and the analysis requiring $\eta$ be sufficiently large? What are the consequences for $\eta$ being too large?

* In the empirical evaluation, what exactly is being presented? It claims to present results over 10 independent evaluation runs—does this suggest that there was one learning run, and at regular intervals, the current policy was saved and run 10 times to compute its average return? Or is this the average of 10 independent learning runs?

* While I appreciate the comparison in B.4 where the impact of single- vs. double-Q in WPPG was explored, is there any reason why WPO + double-Q was not tried? From these ablations, this had a very dramatic effect on WPPG's performance, that WPO might similarly be on par with SAC if it were to also maintain two action-value functions?

* In the empirical evaluation, why were the shaded regions chosen to represent one standard deviation? This is a measure of variation and not confidence, the latter of which is more relevant for making claims about differences in performance. While this can be used to compute a standard error, it would be more informative to present the standard error or confidence intervals directly.

---

> ### Author Response · Authors · 2025-11-26
>
> We sincerely appreciate your thorough review, constructive comments and encouraging feedback. We hope our response below and revisions successfully resolve them.
>
> **W1:** Limited empirical evaluation.
>
> We acknowledge that our empirical evaluation focuses on the MuJoCo suite and agree that this is a limitation. As you noted, the primary contribution of this paper is theoretical; the experiments serve as a proof-of-concept to demonstrate the algorithm's effectiveness. Our new Wasserstein policy gradient variant relies solely on the action gradient, analogous to standard Q-value–based policy optimization methods. As such, we choose MuJoCo continuous-control suite, the standard benchmark for related algorithms (including WPO [1] and SAC). Our selected tasks cover sparse-reward control (Reacher), high-dimensional locomotion (Humanoid), and medium-complexity environments (HalfCheetah, Hopper, Walker2d), which provide a robust testbed for continuous control.
>
> To further strengthen our empirical results, we added a new experiment highlighting the advantages of our framework's support for implicit policies. Following the setup in WPO [1], we constructed a multi-Humanoid environment by combining several tasks in parallel (the revised Appendix B.4): the agent receives concatenated states and outputs concatenated actions jointly. In this setting, our WPPG-I algorithm significantly outperforms SAC, demonstrating the benefits of an expressive implicit policy for complex control. While this submission focuses on MuJoCo, extending the evaluation to other benchmarks is an important direction for future work. We appreciate your understanding!
>
> [1] Pfau, David, et al. "Wasserstein Policy Optimization." Forty-second International Conference on Machine Learning.
>
> **W2 & Q1:** Conflict between the ideal derivation and the analysis.
>
> This is indeed a sharp observation, and we acknowledge the underlying tension. While a small $\eta$ is required for the convergence of the parametrized policy, this assumption is not needed for the ideal policy gradient on the Wasserstein distribution space. Alternatively, by pursuing a distinct analysis route based on Fokker–Planck dynamics and mean-field theory, one can establish convergence without imposing a large $\eta$.
>
> Here, we only sketch the proof. The Wasserstein flow of the policy satisfies
> $$
> \partial_t \pi_t(a \mid s)
> = - \nabla_a \cdot \left( \pi_t(a \mid s)\,\nabla_a Q^{\pi_t}(s,a) \right) + \tau \Delta_a \pi_t(a \mid s),
> $$
> where $\nabla_a$ and $\Delta_a$ denote the gradient and Laplacian with respect to the action variable $a$.
> Differentiating the objective along this flow, we obtain
> $$
> \frac{d}{dt} J(\pi_t)
> = \int d^{\pi_t}(s) \int \pi_t(a \mid s)
> \left\lVert \nabla_a Q^{\pi_t}(s,a)
>       - \tau \nabla_a \ln \pi_t(a \mid s) \right\rVert^2
>  da ds.
> $$
> Therefore, $J(\pi_t)$ is non-decreasing along the Wasserstein flow. Combining this with the performance-difference lemma and using a similar argument to [2] yields convergence.
>
> This reconciles the conflict, albeit at the expense of a more conservative geometric convergence rate. Because this approach diverges substantially from our current analytical framework, we leave it as an intriguing direction for future work.
> Nevertheless, we believe our current result serves as a foundational step for global convergence of Wasserstein-type policy gradients in continuous control.
>
> [2] Chizat, Lénaïc. "Mean-Field Langevin Dynamics: Exponential Convergence and Annealing." Transactions on Machine Learning Research (2022).
>
> **Q2 & Q4:** Empirical evaluation's presentation.
>
> Thank you for pointing out this ambiguity. In the original experiments, each learning curve is based on a single training run. At each evaluation point, we save the current policy and evaluate it over 10 rollouts of the same policy to compute the mean return. We have adjusted the shaded region in our paper to show the 95% confidence interval of the estimated mean instead of the standard deviation.
>
> We also recognize that averaging over multiple independent training runs provides a stronger assessment of an algorithm’s robustness. In the revised Appendix B.4, we include results from multiple training runs with different random seeds and report minimum and maximum returns with shaded regions. These additional results align with the trends reported in the main text and provide more statistically informative performance comparisons.
>
> **Q3:** WPO + double-Q.
>
> Thank you for catching this! We agree that exploring a double-Q variant of WPO is an interesting direction and could potentially improve performance. We did not pursue this variant because we followed the original WPO paper and aimed for a faithful comparison; evaluating WPO in its original form avoids introducing modifications that were not part of the published algorithm. Notably, under the settings used in the WPO paper, the reported performance already exceeds that of SAC even without a double-Q formulation.

---

### Official Review · Reviewer_AEty · 2025-11-02

**Soundness:** 3
**Presentation:** 1
**Contribution:** 3
**Rating:** 4
**Confidence:** 2

**Summary:**

The paper proposes a projected wasserstein proximal policy gradient method and analyzed its global convergence behavior under various assumptions on the policy class. Numerical experiments accompanied the theoretical framework proposed.

**Strengths:**

I find the motivation to use Wasserstein metric in the proximal policy gradient update to be convincing.

**Weaknesses:**

Main concerns:

- Related work: Proximal methods can be interepreted as soft-version of trust region methods. Wasserstein trust region policy optimization methods are considered already a few years back in Terpin, A., Lanzetti, N., Yardim, B., Dorfler, F., & Ramponi, G. (2022). Trust region policy optimization with optimal transport discrepancies: Duality and algorithm for continuous actions. Advances in Neural Information Processing Systems, 35, 19786-19797. Please discuss and compare with this related work.

- Assumptions: there are quite a few assumptions on the coverage and smoothness of policy class. While there are some accompanying discussions next to the assumptions, I am wondering if the authors can provide a concrete example of policy class parametrization therein that satisfy all of the assumptions without vague wording like "designing the neural network to have sufficient smoothness and non-degeneracy."

Minor points:

- There seems to be some latex issues when composing the theorem environment. Instead of showing the theorem environment, it shows plain text with [linear convergence]. Please fix these.

- Line 267: "assuption"

**Questions:**

See previous section

---

> ### Author Response · Authors · 2025-11-26
>
> Thank you for your valuable feedback! We have carefully revised the manuscript to fix the LaTeX issues and typos. More detailed responses to the weakness are as below.
>
> **W1:** Related work Terpin et al. (2022)
>
> Thank you so much for highlighting this reference, and we apologize for the oversight! We have added a dedicated discussion to the literature section. The main differences are summarized below:
>
> *Algorithmically*: At each iteration, Terpin et al. (2022) requires solving an inner subproblem $\max_{a'\in A}\{\hat A^{\pi_{\theta_t}}(s,a')-\lambda c(a,a')\}$ for each action $a$ (and each state $s$), along with an outer minimization over the dual multiplier $\lambda$. Solving this inner subproblem is computationally costly and often intractable due to the nonconvexity of the advantage function with respect to the action (common with neural policies); this can lead to uncontrollable bias in the update. In contrast, our algorithm avoids such inner/outer optimization and relies solely on gradient calculation, making it significantly more tractable and efficient.
>
> *Theoretically*: Terpin et al. (2022) explicitly leaves convergence analysis for future work. In contrast, a primary theoretical contribution of our work is proving the global linear convergence with both exact and inexact $Q$-functions. We would like to note that for continuous action spaces, there has been no provably convergent guarantee prior to our work, despite the fact that optimal-transport-based policy gradient methods have been proposed in various forms.
>
> **W2:** Assumptions.
>
> Thanks for catching this. In the revision, we have relaxed our assumptions as follows: we have completely removed the original Assumption 1(i) regarding the uniformly lower boundedness of the log-density Hessian. Moreover, in Remark 1, we show that Assumption 1(ii) (formerly Assumption 1(iii)) regarding bounded actions can be replaced by uniform boundedness of the policies' second moments.
>
> Regarding your comment, the phrase "designing the neural network to have sufficient smoothness and non-degeneracy" was intended as a sufficient condition for the original Assumption 1(i) (not the original Assumption 1(iv) -- we apologize for the typo). Since Assumption 1(i) has been removed, this condition is no longer required.
>
> Below, we give examples satisfying the revised assumptions.
>
> 1. *Implicit Lipschitz Policies*. Consider policies defined by the pushforward of a standard Gaussian through an $L$-Lipschitz neural network, which are widely studied (e.g., [1,2,3]).
> Since the standard Gaussian satisfies a Talagrand $T_2(1)$ inequality [1], its pushforward under an $L$-Lipschitz map satisfies $T_2(1/L^2)$
> [1]. This verifies Assumption 1(i).
> Furthermore, because the second moment of the standard Gaussian is finite, the second moment of the pushforward distribution is bounded by a constant depending only on $L$, which verifies Assumption 1(ii).
> Finally, Assumption 1(iii) is a universal-approximation type assumption common in theoretical RL literature, and Lipschitz networks possess this universality [2,3]
>
> *Refined result*. For convex, bounded action spaces, we obtain a stronger result (see the new Appendix C): if the initial policy has a continuous density with a uniform upper bound and a strictly positive lower bound on the action space, then Assumption 1(i) holds because the transport maps of all iterates have uniformly bounded Lipschitz norms. Assumption 1(ii) holds trivially, and Assumption 1(iii) holds because Lipschitz networks can approximate any Lipschitz function  and hence all such transport maps, i.e., all the policies.
>
> 2. Mixture-of-Gaussians policies. Assumption 1(i) holds under mild conditions for mixtures of Gaussians [4]; Assumption 1(ii) holds when the component means and covariances are uniformly bounded. Assumption 1(iii) holds for suitable environments where the optimal policy can be approximated by a mixture of Gaussian.
>
> **References**
>
> [1] Michel Ledoux. The concentration of measure phenomenon. Number 89. American Mathematical Society, 2001.
> [2] Cem Anil, James Lucas, Roger Grosse. Sorting Out Lipschitz Function Approximation. Proceedings of the 36th International Conference on Machine Learning (ICML).
> [3] Davide Murari, Takashi Furuya, Carola-Bibiane Schönlieb. Approximation theory for 1-Lipschitz ResNets. arXiv preprint arXiv:2505.12003.
> [4] Chen, Hong-Bin, Sinho Chewi, and Jonathan Niles-Weed. "Dimension-free log-Sobolev inequalities for mixture distributions." Journal of Functional Analysis 281.11 (2021): 109236.

---

### Author Response · Authors · 2025-12-01

**We would like to start by expressing our sincere appreciation for our Area Chair, especially in light of the unexpected incident that has increased the workload. We also thank the reviewers for their careful reading and insightful comments. To make our responses easy to follow, we summarize the main points of our rebuttal below:**

**Across the reviews, the strengths of our work are consistently acknowledged.**

- Reviewer oEc8 and Reviewer qGJK agree that the paper addresses an important and timely problem in policy optimization under Wasserstein geometry, and highlight the strength of the theoretical development as well as the practicality of the algorithm.
- Reviewers AEty and oEc8 agree that the motivation for using the Wasserstein metric in proximal policy updates is well grounded and provides a convincing basis for designing new algorithms.
- Reviewer oEc8, Reviewer qGJK, and Reviewer zK6R explicitly highlight as a key strength that our algorithm natively optimizes general implicit policy models, going well beyond the Gaussian and mixture-of-Gaussians parameterizations typically required by existing proximal and actor–critic methods.


Taken together, these assessments summarize our contributions: we present one of the very first global convergence analyses of a Wasserstein-type policy optimization method in continuous action spaces. In fact, our analysis resolves an open convergence question left by previous Wasserstein policy optimization methods, such as OT-TRPO (Terpin et al., 2022 [1]). We also propose an efficient algorithm that operates directly in Wasserstein geometry, uses only action-value gradients, does not require policy log-densities, and thus supports expressive implicit policies while achieving competitive performance on standard continuous-control benchmarks.

**The main concerns are:**

(1) **About assumptions and their realizability (Reviewer AEty, Reviewer qGJK).**
The main concern is that our assumptions seem strong and require checking that each assumption is realizable in practice.
We addressed this by:
- simplifying the assumptions, removing the Hessian lower-bound condition on the policy potential and replacing bounded action support with the milder requirement of uniformly bounded second moments;
- providing concrete, verifiable examples of policy classes (Gaussian base distributions pushed forward through Lipschitz neural networks, and mixtures of Gaussians) that satisfy the required $T_2$ transport–entropy condition;
- clarifying that many standard distributions (Gaussian, uniform, mixtures) satisfy $T_2$, and that $T_2$ is preserved under Lipschitz maps, which covers typical neural-network policies;
- adding a refined bounded-action analysis (Appendix C) showing that, in the case of bounded action spaces (quite common in practice), all policy iterates produced by WPPG preserve $T_2$, making the assumption both natural and stable along the learning trajectory.

(2) **About empirical evaluation, robustness, and computational overhead compared with the previous algorithm WPO (Reviewer oEc8, Reviewer qGJK).**
The concerns are that the empirical evaluation focuses on MuJoCo and that computational cost and robustness are not fully explained.
We addressed this by:
- adding a more challenging multi-Humanoid experiment and extended ablations (over step size, entropy coefficient, sample number, and latent dimension), where the implicit variant WPPG-I achieves competitive and in some cases superior performance compared to SAC (SOTA), illustrating the benefit of expressive implicit policies;
- clarifying our evaluation protocol (how training and evaluation runs are averaged) and updating plots to use confidence intervals;
- explaining that WPPG relies solely on the action gradient, analogous to standard Q-value–based policy optimization methods, and, unlike WPO, it does not compute the gradient of the policy's log-density, thereby reducing computational overhead. Moreover, WPPG supports richer implicit policies.

---

> ### Author Response · Authors · 2025-12-01
>
> (3) **About novelty (Reviewer zK6R).**
> The concerns are whether the theory is essentially a restatement of existing KL-based results (e.g., Lan [2]) or finite-state, finite-action results (e.g., Song et al. [3]), and whether the analysis is limited to tabular/oracle updates.
> We addressed this by:
> - clarifying that Lan’s linear-rate mirror-descent analysis relies on the KL three-point identity, which does not hold in Wasserstein space, so our proof develops new descent and contraction inequalities tailored to Wasserstein geometry, yielding (to our knowledge) one of the very first global linear convergence results for Wasserstein-based policy optimization;
> - explaining that Song et al. work only with finite-action, finite-state tabular policies and do not exploit any property of Wasserstein geometry itself beyond an action-space diameter bound (in fact, their analysis can be applied to any bounded metric), whereas our Lemma 7 and Theorems 1–2 use the specific structure of Wasserstein geometry to obtain a nontrivial contraction;
> - making explicit that, although the oracle updates are per-state, our analysis is not restricted to the tabular case: we introduce a standard realizability error $\delta$ in RL theory literature to quantify the projection from oracle distributions to parametric policies and show that it contributes only an $O(\delta)$ term to the global convergence. We further extend the result to inexact critics, establishing convergence in the approximate Q-function setting.
>
> **References**
>
> [1] Terpin, A., Lanzetti, N., Yardim, B., Dörfler, F., & Ramponi, G. (2022). Trust region policy optimization with optimal transport discrepancies: Duality and algorithm for continuous actions. Advances in Neural Information Processing Systems, 35, 19786–19797.
>
> [2] Guanghui Lan. Policy mirror descent for reinforcement learning: Linear convergence, new sampling complexity, and generalized problem classes. Mathematical Programming, 198(1), 1059–1106, 2023.
>
> [3] Song, Jun, et al. Provably Convergent Policy Optimization via Metric-aware Trust Region Methods. Transactions on Machine Learning Research, 2023.

---

### Meta-Review · Area_Chair_Si6u · 2026-01-09

**Summary:**

The authors present a Wasserstein proximal gradient algorithm that can be effectively approximated by a family of parametric policies defined implicitly through the reparameterization trick. The resulting algorithm closely resembles a regularized form of SAC or DDPG - essentially, DDPG optimized via stochastic gradient Langevin dynamics rather than SGD (or, SGD with a small amount of Gaussian noise added to the parameters). The authors claim that one advantage is that it can be used with policies that have no explicit action probability and instead are defined by reparameterization, but this seems backwards - similar methods that take the gradient of the Q function to update the policy like SAC and DDPG also work with reparameterized policies, only methods like WPO can work with action distributions with discrete latent variables, which cannot be parameterized by pushing a fixed distribution forward. The math is thorough and rigorous. The reviewers had some concerns about the novelty of the results, but the authors clarified the difference from existing work. Empirical results were interesting, but the main contribution of the paper is theoretical. While the authors addressed how their result differs from specific papers raised by the reviewers, I remain concerned that the main result is not particularly novel, given how minor the difference is from standard SAC or DDPG. For instance, a quick search for "stochastic gradient Langevin dynamics" and "deterministic policy gradient" yields this 2019 workshop paper which studies a nearly identical policy update for continuous control: https://optrl2019.github.io/assets/accepted_papers/36.pdf. I believe the authors should do more to clarify the novelty of the result - both the theoretical novelty, but also how the exact policy update differs from existing methods like SAC and DDPG.

**Reviewer Concerns:**

Many reviewers were unclear on how novel the results were. The authors clarified this.

**Reviewer Scores:**

It is difficult to say. Most reviewers were weakly against acceptance, and I do not know if enough would change their mind to justify acceptance.

---

### Decision · Program_Chairs · 2026-01-26

Reject